

# Thermodynamic Properties of Seawater, Ice and Humid Air: TEOS-10, Before and Beyond

Rainer Feistel[1]

[1]Leibniz Institute for Baltic Sea Research (IOW), Warnemünde, D-18119, Germany

*Correspondence to*: Rainer Feistel (Rainer.Feistel@IO-Warnemuende.de)

**Abstract.** In the terrestrial climate system, water is a key player in the form of its different ambient phases of ice, liquid and vapour, admixed with sea salt in the ocean and with dry air in the atmosphere. For proper balances of climatic energy and entropy fluxes in models and observation, a highly accurate, consistent and comprehensive thermodynamic standard framework is requisite in geophysics and climate research. The new "Thermodynamic Equation of Seawater – 2010"

(TEOS-10) constitutes such a standard for properties of water in its various manifestations in the hydrological cycle. TEOS-10 has been recommended internationally in 2009 by the Intergovernmental Oceanographic Commission (IOC) to replace the previous 1980 seawater standard, EOS-80, and in 2011 by the International Union of Geodesy and Geophysics (IUGG) "as the official description for the properties of seawater, of ice and of humid air". This paper briefly reviews the development of TEOS-10, its novel axiomatic properties, new oceanographic tools it offers, and important tasks that still

await solutions by ongoing research. Among the latter are new definitions and measurement standards for seawater salinity and pH, in order to establish their metrological traceability to the International System of Units (SI), for the first time after a century of widespread use. Of similar climatological relevance is the development and recommendation of a uniform standard definition of atmospheric relative humidity that is unambiguous and rigorously based on physical principles.

*The leading thermodynamic properties of a fluid are determined by the relations*

*which exist between volume, pressure, temperature, energy, and entropy…*

*But all the relations existing between these five quantities for any substance …*

*may be deduced from the single relation*

*existing for that substance between volume, energy, and entropy.*

*Josiah Willard Gibbs, 1873b*



# 1 Introduction

In the context of recent global warming and the anthropogenic greenhouse effect of carbon dioxide ($CO_2$), the pivotal article of Svante Arrhenius (1896) found public attention well beyond the scientific communities of meteorologists and climatologists. Much less known, however, is the lecture of Heinrich Hertz given in 1885 in which he analysed the thermodynamics of the hydrological cycle in the climate system as a "gigantic steam engine" (Mulligan and Hertz, 1997, p. 41). In fact, rather than $CO_2$, water in the troposphere in the form of humidity and clouds contributes the major part to the overall greenhouse effect (Abbot and Fowle, 1908; Emden, 1913; Trenberth et al., 2007; Lacis et al., 2010; Schmidt et al., 2010; Feistel and Ebeling, 2011; Feistel 2015, 2017; Lovell-Smith et al., 2016). The global water cycle, its observation and modelling, poses a fundamental challenge for climate research (Sherwood et al., 2010; Reid and Valdés, 2011; Tollefsen, 2012; Fasullo and Trenberth, 2012; Josey et al., 2013; Stevens and Bony, 2013; IPCC, 2013).

Saline water in the oceans, freshwater lakes and rivers, polar ice caps, humid air and clouds form a highly dynamic, coupled system of water in different phases and mixtures. Densities, heat capacities and the latent heats of mutual phase transitions of water play a key role for the transformation and distribution processes of energy between initial absorption of incoming solar radiation and final export of thermal radiation. A comprehensive, consistent quantitative knowledge of water properties in its various appearances is requisite for the analysis of measurements and the development of numerical models. Moreover, climate research is carried out at various places of the world and extends over timespans of several human generations. Mutual comparability of research results is indispensable, which rigorously requires that well-defined, highly accurate international standards must be applied, such as the International System of Units (SI).

TEOS-10, the *Thermodynamic Equation of Seawater – 2010* constitutes such an international standard for the thermodynamic properties of water in the climate system. In addition to seawater in particular, it also covers the properties of ice and humid air in a perfectly consistent, comprehensive way with unprecedentedly high accuracy. By the numerical coefficients of its empirical thermodynamic potentials, TEOS-10 represents in a mathematically most compact way the results of an enormous amount of very different experimental studies of water, ice, seawater and air properties. TEOS-10 has been recommended internationally in 2009 by the Intergovernmental Oceanographic Commission (IOC) to replace the previous 1980 seawater standard, EOS-80, and in 2011 by the International Union of Geodesy and Geophysics (IUGG) "as the official description for the properties of seawater, of ice and of humid air". Beginning in 2006, TEOS-10 was developed in close cooperation between the SCOR/IAPSO Working Group 127 on Thermodynamics and Equation of State of Seawater, and the International Association for the Properties of Water and Steam (IAPWS). TEOS-10 is officially defined by its manual (IOC et al., 2010) and is supported by open-source software libraries (available from the internet at the TEOS-10 homepage, www.teos-10.org), several open-access IAPWS documents (IAPWS AN6-16, 2016) as well as various supporting scientific articles, such as the definition of the new Reference-Composition Salinity Scale (Millero et al., 2008) or a new





thermodynamic definition of relative humidity (Feistel and Lovell-Smith, 2017). A collection of background articles of TEOS-10 was published as a Special Issue of Ocean Science (Feistel et al., 2008b).

This paper is organised as follows. Thermodynamic potentials are not necessarily taught in courses of oceanography or other geosciences, so Section 2 provides a short introduction to this theoretical method developed about 150 years ago. The most convenient mathematical formalism of manipulating partial derivatives with different sets of variables, as necessary for the work with those potentials, is the Jacobi method explained in Appendix A. Section 3 describes details of the development of the thermodynamic potentials for fluid water, ice, seawater and humid air, which constitute the core of TEOS-10, together with the scales used for temperature and salinity. Section 4 deals with the structure of TEOS-10, its axiomatic approach to the computation of thermodynamic properties, and the digital, open-source libraries provided by TEOS-10. Members of the SCOR/IAPSO Working Group 127, who participated in the annual meetings and successfully completed these demanding tasks within a few years of intense cooperative work, can be found in Appendix B. Section 5 describes three important pending problems that still await solutions beyond TEOS-10.

The relatively large number of references cited in this review includes numerous sources that are of relevance in the context of TEOS-10 and belong to a variety of different scientific disciplines. Albeit, these references do not even include the extensive lists of numerous experimental works that provided the fundamental property data of water, seawater, ice and humid air, ultimately being represented by TEOS-10 in an integrated, compact form.

## 2 Thermodynamic Potentials

A thermodynamic potential is a mathematical function from which all thermodynamic properties of a given many-particle system at equilibrium can be derived using formal thermodynamic rules. The system under study may be a pure substance, such as water or ice, a mixture such as seawater or humid air, or a multi-phase composite such as sea ice (ice with brine pockets) or a cloud (air with liquid droplets). J. Willard Gibbs discovered the existence of such functions already in 1873 (Gibbs, 1873a,b), but their practical use remained limited before powerful computers became commonplace in science. In contrast, thermodynamic potentials have always played a key role in theoretical physics (Landau and Lifschitz, 1966). The partition functions of statistical thermodynamics are mathematical expressions for the Boltzmann entropy, the Helmholtz potential and the Landau potential, respectively, obtained from the micro-canonical, canonical and grand-canonical ensembles.

Depending on which external conditions are imposed on a given sample, an associated potential function needs to be chosen. For example, if we know the temperature $T$, the volume $V$ and the total mass $m$, the potential to be used is the *Helmholtz energy*, $F(m, T, V)$, or similarly, as a function of temperature and density, $\rho$, the *specific Helmholtz energy*, $f(T, \rho) = F/m$. A

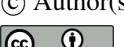



typical case of such conditions is an experimental setup of a liquid or gas in a given rigid container, in order to measure the pressure or the isochoric heat capacity at different temperatures or densities. In geophysics, such as in the atmosphere or in the ocean, typically the pressure $p$ rather than the volume of a fluid parcel is available. Then, the *Gibbs energy G(m, T, p)*, or similarly, the *specific Gibbs energy*, $g(T, p) = G/m$, is the appropriate potential from which properties of interest, such as the

density, entropy or enthalpy of the parcel, can be computed. For brevity, the terms *Gibbs function* and *Helmholtz function*, respectively, are used in this paper for the specific Gibbs and Helmholtz energies expressed in terms of their particular natural independent variables. The two potentials may be converted into each other, if either $f(T, \rho)$ is known, by

$$g(T,p) = f + \rho \frac{\partial f}{\partial \rho}, \qquad\qquad p = \rho^2 \frac{\partial f}{\partial \rho}. \qquad\qquad (1)$$

or, if $g(T, p)$ is given, by

$$f(T,\rho) = g - p \frac{\partial g}{\partial p}, \qquad\qquad \rho^{-1} = \frac{\partial g}{\partial p}. \qquad\qquad (2)$$

Transformation equations of this kind between different sets of independent variables are commonly known as *Legendre transforms* (Margenau and Murphy, 1943; Landau and Lifschitz, 1966; Alberty, 2001). Another potential function of significant interest in TEOS-10 is the specific enthalpy $h(\eta, p)$ as a function of the specific entropy, $\eta$, and the pressure. It is derived from a TEOS-10 Gibbs function $g(T, p)$ by the Legendre transform

$$h(\eta,p) = g - T \frac{\partial g}{\partial T}, \qquad\qquad \eta = -\frac{\partial g}{\partial T}, \qquad\qquad (3)$$

and from a TEOS-10 Helmholtz function $f(T, \rho)$ by

$$h(\eta,p) = f - T \frac{\partial f}{\partial T} + \rho \frac{\partial f}{\partial \rho}, \qquad\qquad \eta = -\frac{\partial f}{\partial T}, \qquad\qquad p = \rho^2 \frac{\partial f}{\partial \rho}. \qquad (4)$$

The partial derivatives are understood here as being taken at constant values of the respective other, so-called „natural" independent variables associated with a given potential.

Potential functions, such as $F$, $f$, $G$, $g$ or $h$ above, may also be expressed in terms of alternative independent variables, but then they may lose their capabilities as potential functions from which all other quantities may be derived. For example, entropy can only incompletely be computed from the function $h(T, p)$. The "original" thermodynamic potential studied by Gibbs, see his initial quotation, was internal energy as a function of volume and entropy; this function is rarely used in

geophysics or engineering, mainly because its input parameter entropy cannot be measured experimentally (Tillner-Roth, 1998). Among the most relevant measurable properties derived from the potential functions above are density, $\rho$,

$$\frac{1}{\rho} = \frac{V}{m} = \left(\frac{\partial g}{\partial p}\right)_T = \left(\frac{\partial h}{\partial p}\right)_\eta, \qquad\qquad (5)$$

isobaric specific heat capacity, $c_p$,

$$c_p = \left(\frac{\partial h}{\partial T}\right)_p = T \left(\frac{\partial \eta}{\partial T}\right)_p = -T \left(\frac{\partial^2 g}{\partial T^2}\right)_p, \qquad\qquad (6)$$

sound speed, $c$,



$$\frac{1}{c^2} = \left(\frac{\partial \rho}{\partial p}\right)_\eta = -\varrho^2 \left(\frac{\partial^2 h}{\partial p^2}\right)_\eta = \varrho^2 \frac{\left(\frac{\partial^2 g}{\partial p \partial T}\right)^2 - \left(\frac{\partial^2 g}{\partial T^2}\right)_p \left(\frac{\partial^2 g}{\partial p^2}\right)_T}{\left(\frac{\partial^2 g}{\partial T^2}\right)_p}, \tag{7}$$

and the adiabatic lapse rate, $\Gamma$,

$$\Gamma = \left(\frac{\partial T}{\partial p}\right)_\eta = \frac{\partial^2 h}{\partial \eta \partial p} = \frac{\frac{\partial^2 g}{\partial T \partial p}}{\left(\frac{\partial^2 g}{\partial T^2}\right)_p}. \tag{8}$$

Here, for uniqueness of the mathematical expressions, the subscripts at the brackets indicate the variables that are kept

constant when the partial derivative is carried out. Extended lists of such thermodynamic relations for the various quantities implemented in TEOS-10 are available from the manual (IOC et al., 2010) and from the digital supplements of Feistel et al. (2010a) and Wright et al. (2010a). The conversion of formulas from one set of independent variables to another is done using the chain rule of differential calculus. In a convenient multi-variable version, this rule is known as the *Jacobi method*. Appendix A demonstrates this method tutorially and how it is applied to derive formulas like (5)-(8) in a straightforward

manner, such as eq. (A.18) for the sound speed.

In order to quantitatively determine a thermodynamic potential of a given substance of interest, a suitable but largely arbitrary mathematical expression for the potential function is designed by trial and error. The function depends on a set of $N$ adjustable parameters, say $\boldsymbol{a} = (a_1, \dots a_N)$, which are estimated by simultaneous numerical regression with respect to all

reliable experimental data available for the properties (5) – (8) and others. For each substance involved, two of those parameters remain unknown, representing the absolute energy and absolute entropy of that substance, unavailable from thermodynamic measurements. These parameters are usually specified by conventionally assuming certain values to selected quantities at a chosen reference state. In TEOS-10, internal energy and entropy of liquid water are assumed to take zero values at the common solid-liquid-gas triple point. Similar assumptions were also employed for dry air and dissolved sea

salt, see Section 3. For consistency, it is important to implement for any given substance the same conventional values in any mixture or phase where it is present, such as for the water substance $H_2O$ in ice, in seawater and in humid air (Feistel et al., 2008a).

## 3 Fundamental Thermodynamic Potentials of TEOS-10

### 3.1 Temperature Scales

For the measurement of temperatures, temperature-dependent properties of certain materials are frequently used, such as the volume expansion of liquid mercury or the electrical resistance of platinum. Sensors of this kind need to be calibrated, that is, a quantitative relation between that property and selected numerical temperature values must be specified. For the currently valid International Temperature Scale of 1990 (ITS-90), "the unit of the fundamental physical quantity known as thermodynamic temperature, symbol $T$, is the kelvin, symbol K, defined as the fraction 1/273.16 of the thermodynamic



temperature of the triple point of water" (Preston-Thomas, 1990). In practice, many published geophysical property equations are expressed in terms of earlier, meanwhile obsolete scales, such as the International Practical Temperature Scale of 1968, IPTS-68. Among those obsolete equations are, in particular, the International Equation of State of Seawater, EOS-80, and the Practical Salinity Scale of Seawater, PSS-78 (Unesco, 1981a,b, 1983).

TEOS-10 is expressed in terms of ITS-90. Experimental data were converted from their original values according to the transformation rules published by Rusby (1991), Goldberg and Weir (1992), Rusby et al. (1994), and Weir and Goldberg (1996). In the vicinity of the triple point of water, the deviations between different temperature scales are generally small, in particular for oceanographic applications (Saunders, 1990; Feistel and Hagen, 1995; Feistel, 2008a). ITS-90 is expected to
remain in practical use for the foreseeable future.

On 20 May 2019, the World Metrology Day, a so-called "new SI" is expected to become formally introduced at the International Bureau of Weights and Measures (BIPM) in Sèvres near Paris. Along with it, a new thermodynamic temperature scale will be established in 2018 (Fellmuth et al., 2016; PTB, 2017; CCT, 2017; CGPM, 2018). This new
temperature $T$ and the ITS-90 temperature $T_{90}$ will be considered as two different physical quantities expressed in the same unit, namely the new kelvin (rather than being the same physical quantity expressed in two different units, the K and a fictitious "$K_{90}$"). The definition of the kelvin contained in the 1990 definition will be superseded by the new kelvin definition in terms of the Boltzmann constant and the joule (CGPM, 2018). In ITS-90, the triple-point temperature of water will remain at exactly $T_{90} = 273.16$ K. This value will also be valid for the thermodynamic temperature but only within an uncertainty yet
to be specified, likely about 160 µK. By definition, the Boltzmann constant, $k = 1.380\,649 \cdot 10^{-23}$ J K$^{-1}$, the Avogadro constant, $N_A = 6.022\,140\,76 \cdot 10^{23}$ mol$^{-1}$, and the molar gas constant, $R = k \cdot N_A$, will take eternally fixed, exact values in the new SI (Fischer, 2016; PTB, 2017). This is a very practical future aspect as the various IAPWS documents supporting TEOS-10 implement several slightly different values of the gas constant, according to varying official values it took at the time the particular document was developed (IAPWS G5-01, 2016).

**3.2 Helmholtz Function of Fluid Water**

The joint Helmholtz function, $f^F(T, \rho)$, of fluid water published by Wagner and Pruß (2002), also known as the IAPWS-95 equation, is identical with the Helmholtz functions of liquid water, $f^F \equiv f^W$, and water vapour, $f^F \equiv f^V$, and represents a key constituent of TEOS-10. In compact form, the latest version of this equation is available from IAPWS R6-95 (2016). Since its first release in 1995, the latter document has undergone several minor revisions, however, without affecting the calculated
values of any measurable properties. The equation is also available from several other references, such as IOC et al. (2010) and Feistel et al. (2010b). The formulation is valid in the entire stable fluid region of H$_2$O from the melting curve to 1273 K at pressures up to 1000 MPa; the lowest temperature on the melting curve is 251.165 K at 208.566 MPa (Wagner et al., 2011; IAPWS R14-08, 2011), see Fig. 1. As an additional part of TEOS-10, a low-temperature extension of the IAPWS-95



equation for water vapour down to 50 K is given by Feistel et al. (2010b) and IAPWS G9-12 (2012). IAPWS-95 is valid for air-free water with an isotopic composition of Vienna Standard Mean Ocean Water, VSMOW (IAPWS G5-01, 2016).

Helmholtz functions are preferred description tools for fluids because they may cover the wide density range from the gas to the liquid phase by a single, unique formula. In contrast, Gibbs functions take multiple values in the $T$-$p$ vicinity of the saturation curve where liquid and gas coexist, so that in practice two separate Gibbs functions, $g^W$ and $g^V$, respectively, are specified for liquid and vapour. Consistent with TEOS-10, there is an equation for the Gibbs function for liquid water under oceanographic conditions (Feistel, 2003; IAPWS SR7-09, 2009), an equation for the Gibbs function for liquid water at atmospheric pressure (Hrubý et al., 2009; IAPWS SR6-08, 2011), and a virial expansion of the Gibbs function for water

vapour (Feistel et al., 2015). Recently, a Gibbs function for supercooled, metastable liquid water has also been developed (IAPWS G12-15) which is not included in TEOS-10.

For a pure substance, the chemical potential equals its Gibbs function. Consequently, along the saturation curve, $p^{sat}(T)$, in the $(T, p)$ diagram from the common triple point to the critical point of water, see Fig. 1, the Gibbs function of liquid water

equals the Gibbs function of water vapour, $g^W(T, p^{sat}) = g^V(T, p^{sat})$. From this equilibrium condition, the saturation pressure as a function of the temperature, $p^{sat}(T)$, can be calculated iteratively. A simple analytical approximation formula of this solution is available (IAPWS SR1-86, 1992).

In their common ranges of validity, IAPWS-95 is consistent with the most accurate metrological equation of state of liquid

water, the so-called CIPM-2001 equation (Tanaka et al., 2001; Harvey et al., 2009; IAPWS AN4-09, 2009). This means that under ambient conditions, density is known within an uncertainty of 1 ppm. This accuracy permits the use of IAPWS-95 for the calibration of high-precision density meters (Wolf, 2008). By means of such instruments, seawater salinity may be determined from measured densities, and sea-salt composition anomalies may be detected (Millero et al., 2008; Feistel et al., 2010c; Wright et al., 2011; Woosley et al., 2014; Pawlowicz et al., 2016; Schmidt et al., 2016, 2018; Budéus, 2018), see also

Section 3.4.

In TEOS-10, the IAPWS-95 equation replaced the earlier equations of state of liquid water of Bigg (1967) and Kell (1975), on which the former seawater standard EOS-80 was based (Unesco, 1981b). This change of the pure-water equation made it possible to resolve systematic problems previously encountered with the sound speed of seawater at high pressures (Dushaw

et al., 1993; Millero and Li, 1994; Feistel, 2003).

**3.3 Gibbs Function of Ice Ih**

Hexagonal ice I is the only stable solid phase of water in the terrestrial atmosphere, hydrosphere and cryosphere. Several textbooks on ice were published in the past, such as by Dorsey (1968), Hobbs (1974) or Petrenko and Whitworth (1999),



offering separate empirical equations for selected properties. The first Gibbs functions for ice Ih were developed by Feistel and Hagen (1995, 1998) and Tillner-Roth (1998), valid in the vicinity of the melting curve. Including numerous additional data, the two approaches were combined by Feistel and Wagner (2004, 2005, 2006, 2007; IAPWS R10-06, 2009) for an IAPWS equation for ice Ih that covers a wide range on temperatures and pressures and is significantly more accurate than

several single-property equations published before. Theoretically, a Helmholtz function for ice Ih can smoothly be combined with that for fluid water into a single 3-phase Helmholtz function for water, valid everywhere near the triple point, but such an attempt has not yet been made.

Measurement of ice properties is experimentally challenging in particular for sluggish aging effects; if ice is brought under

different temperature or pressure conditions, the establishment of the related new equilibrium crystal structure may take hours or even days till completion. Consequently, measurement series conducted too quickly may suffer from the risk of systematic hysteresis effects. Published data on ice compressibility vary up to a factor of 3.6 between different authors. In TEOS-10, the related uncertainty of mechanical measurements could be reduced by a factor of 100 to about 1 % (Feistel and Wagner, 2004, 2005) using crystallographic spectroscopy data (Gammon et al., 1980, 1983; Gagnon et al., 1988), see Fig. 2.

The chemical potential of a pure substance equals its Gibbs function. Consequently, along the freezing curve in the ($T$, $p$) diagram, Fig. 1, the Gibbs function of ice Ih equals the Gibbs function of liquid water, and along the sublimation curve, the Gibbs function of ice Ih equals the Gibbs function of water vapour. From these equilibrium conditions, the freezing pressure and the sublimation pressure as functions of the temperature can be calculated iteratively. Convenient analytical approximation formulae are available for both curves (Wagner et al., 2011; IAPWS R14-08, 2011).

It is well-known that the latent heats of evaporating or freezing water possess exceptionally large values, and that this fact plays an important role in the climate system. The latent heat of sublimation is less familiar, even though it amounts to the sum of the latent heats of melting and evaporation. Air is dry under the polar high-pressure cells of the troposphere, and sublimation rates of glaciers, icebergs or polar ice caps are significant there. Antarctic dry valleys are famous for their

extreme conditions. Sublimation fluxes are approximately driven by relative humidity, which in turn is governed by the sublimation pressure of ice. In TEOS-10, the sublimation pressure can be calculated directly from the thermodynamic potentials of ice and vapour; it turned out that the uncertainty of those computed TEOS-10 values of the sublimation pressure is typically about a factor of 10 smaller than the uncertainty of the most accurate direct measurements, such as those of Fernicola et al. (2012) or Bielska et al. (2013). The limitation of accuracy of the computed values results mainly from the

uncertainty of the heat capacity data of ice and water vapour that entered the thermodynamic potentials (Wagner et al., 2011). Moreover, estimated upper and lower bounds of the heat capacity values of water vapour result directly in corresponding lower and upper bounds for the computed sublimation temperature at given pressure (Feistel and Wagner, 2007). The TEOS-10 equation extends to much lower temperatures than other standard formulas, such as that of Murphy and



Koop (2005). For these reasons, TEOS-10 and its sublimation-pressure equation are preferable for geophysical applications such as models of ice clouds down to temperatures of 80 K (Lübken et al., 2009).

Another distinctive feature of the TEOS-10 ice equation is its exceptional accuracy in ITS-90 freezing temperatures of air-free water, such as 273.152 519 K at atmospheric pressure, 1013.25 hPa, within an uncertainty of just 2 µK (Feistel and Wagner, 2005, 2006; Feistel, 2012, Feistel et al., 2016). For air-saturated water, this freezing point is lowered to the common ice point of 273.150 019 K with an estimated uncertainty of 5 µK (Harvey et al., 2013).

### 3.4 Salinity Scales

Seawater salinity was defined and measured in different ways in the past, and expressed in different units. TEOS-10 is formulated in terms of *Absolute Salinity*, $S_A$, defined as the mass fraction of sea salt dissolved in water, usually expressed in grams per kilogram (IOC et al., 2010). For IAPSO Standard Seawater (SSW), the best estimate available for its Absolute Salinity is the *Reference-Composition Salinity*, $S_R$, for short simply *Reference Salinity*, calculated from the atomic weights of the chemical constituents of sea salt (Millero et al., 2008). Present-day oceanographic measurement devices, however, report values of Practical Salinity, $S_P$, which are computed from the electrical conductivity as defined by the Practical Salinity Scale of 1978, PSS-78 (Unesco 1981a, 1983). Practical Salinity is a unitless quantity, but because there are various other obsolete or non-standard, unitless salinity measures, some oceanographers prefer to attach "psu" (for "practical salinity unit") to reported figures of $S_P$. This inofficial use of "psu" does not cause any harm but has been subject to controversial emotional debates (Unesco 1985, 1986; Millero, 1993). As an aside, a similar discussion is currently going on regarding the use of "%rh" to denote unitless values of relative humidity (RH) expressed in percent (Lovell-Smith et al., 2016) for an easier distinction between RH and, say, specific humidities or relative uncertainties of RH.

For SSW, Reference Salinity is obtained from Practical Salinity by the formula
$$S_R = 35.165\,04 \text{ g kg}^{-1}\, S_P/35 \approx S_P \cdot 1.004\,715 \text{ g kg}^{-1}. \tag{9}$$
For historical data, this equation can only be applied to salinity measurements carried out after 1978. Older legacy data are preferably converted from readings of chorinity, *Cl*, by
$$S_R = 35.165\,04 \cdot 1.806\,55\, Cl/35 \approx 1.815\,069\, Cl. \tag{10}$$
Chlorinity is defined as the mass ratio of grams of pure silver necessary to precipitate the halogens (that is, chlorine, bromine, iodine) from 328.5234 g of seawater (Jacobsen and Knudsen, 1940; Sverdrup, 1942; Millero et al., 2008). By this definition, the chlorinity value is slightly greater than the mass fraction of chlorine in seawater, $w_{Cl} = 0.998\,9041\, Cl$ (Millero et al., 2008). From the 19[th] century on, chlorinity values were rather consistently defined and determined by titration as the most accurate available salinity measure (Forch et al., 1902; Carritt, 1963; Lyman, 1969; Lewis, 1980; Pawlowicz et al., 2016; Burchard et al., 2018). In contrast, the unsatisfactory situation with the definition of salinity before PSS-78 was that "nobody has measured salinity for 60 years. At present chlorinity, conductivity and refractive index measurements are all



being converted to 'salinity' by inadequate tables of often doubtful origin. All make assumptions regarding constancy of relative proportions of the various ions, which are doubtful and may be quite unjustified, to the precision of our modern measurements" (Unesco, 1976, p. 2 therein).

Absolute Salinity as defined in the context of TEOS-10 is the first officially revised measure of the mass fraction of salt dissolved in seawater (McDougall et al., 2008; Pawlowicz et al., 2016) after that of Knudsen and Sørensen more than a century ago (Forch et al., 1902). For seawater of standard composition, the Reference Salinities from eqs. (9) and (10) should result in numerical values that agree with each other within measurement uncertainty. Then, Absolute Salinity $S_A$ is estimated from

$$S_A = S_R \qquad\qquad\qquad (11)$$

to be used as salinity input variable for TEOS-10 equations. However, salinities determined alternatively from both conductivity and chlorinity measurements may produce mutually inconsistent results. This happens especially if the chemical composition of seawater deviates from that of SSW, such as in the Baltic Sea (Wirth, 1940; Kwiecinski, 1965; Rohde, 1966; Millero and Kremling, 1976; Feistel et al., 2010c) or regionally in the global ocean (Millero et al., 1976, 2008,

2009, 2011; Tsunogai et al., 1979; Millero, 2000; Pawlowicz, 2010; Wright et al., 2011; Ushida et al., 2011; McDougall et al., 2012; Woosley et al., 2014).

Before TEOS-10, the international standards attempted to avoid inconsistent results caused by using different methods of salinity determination. With the introduction of PSS-78, the further use of chlorinity titration was discouraged for this reason

(Lewis and Perkin, 1978). TEOS-10 is the first international seawater standard where chemical composition anomalies are explicitly accounted for. In cases when eqs. (9) and (10) do not match, the Reference Salinity (9) is to be corrected by

$$S_A = S_R + \delta S_A, \qquad\qquad\qquad (12)$$

instead of eq. (11), from density measurements or other suitable information, such as for the estimated actual composition. The required correction,

$$\delta S_A = \frac{1}{\beta}\left\{\frac{\rho_{\text{meas}}}{\rho^{\text{SW}}(S_R)} - 1\right\}, \qquad\qquad\qquad (13)$$

may be estimated from the measured density, $\rho_{\text{meas}}$, and the computed TEOS-10 seawater density, $\rho^{\text{SW}}$, evaluated at the Reference Salinity and the same $T$ and $p$, given by eq. (9), see Fig. 3. Here, $\beta$ is the haline contraction coefficient of seawater which has a typical value of about 0.66 at 35 g kg$^{-1}$, atmospheric pressure and 300 K. As a revival of previous, so-called specific-gravity methods (Forch et al., 1902; Krümmel, 1907), this density-based approach received appreciation again for

the high accuracy of modern vibrating-tube density meters (Kremling, 1971; Wolf, 2008; Schmidt et al., 2016, 2018).

While the validity of Practical Salinity is constrained to values of $S_P$ between 2 and 42 (Unesco, 1981c), the new Reference Salinity can be determined also for higher concentrations because it is - by its definition - a physical mass fraction, rather





than a merely formal measure associated with the conductivity. Concentrated brines may be diluted with pure water until the salinity falls in the validity range of PSS-78. Then, $S_P$ is measured, converted to $S_R$ and, using the amount of added water and mass-conserving mixing rules, the Reference Salinity of the original sample is calculated (Millero at al., 2008).

### 3.5 Gibbs Function of Seawater

Thermodynamic properties of a Gibbs function for seawater were first discussed theoretically by Fofonoff (1962). However, in the context of the development of EOS-80, no attempt had been reported of actually constructing such a function numerically. The first practically implemented Gibbs function (Feistel, 1991, 1993; Feistel and Hagen, 1994) was consistent with EOS-80 and, beyond that, consistently provided additional properties such as entropy, enthalpy and the chemical potentials of water and sea salt that had partly been made available elsewhere before (Millero and Leung, 1976; Unesco,

1981b; Millero, 1983).

An improved potential version (Feistel and Hagen, 1995) addressed several weaknesses of its predecessor as well as of EOS-80, such as conversion to the ITS-90 temperature scale. Additional data were included to improve the equation in the vicinity of the temperature of maximum density (Caldwell, 1978; Siedler and Peters, 1986) and for deep-water sound speeds

(Dushaw et al., 1993; Millero and Li, 1994). For a corrected asymptotic zero-salinity limit, the 1995 Gibbs function, $g^{SW}$,

$$g^{SW}(S_P, T, p) = g^{W}(T, p) + g_1(T)S_P \ln S_P + g_2(T, p)S_P + g_3(T, p)S_P^{3/2} + \ldots \qquad (14)$$

recalculated the logarithmic ideal-solution term, $g_1$, and Debye's limiting law of dilute electrolytes, $g_3$, of the salinity expansion, eq. (14), explicitly from an exactly electro-neutral chemical composition model of sea salt. As a precursor of the Reference Composition of TEOS-10 (Millero et al., 2008), this 1995 stoichiometry was a slightly modified version of that

defined by Millero (1982). The new Gibbs function provided a reliable thermodynamic basis for the calculation of novel oceanographic tools such as Conservative Temperature (McDougall 2003), of new efficient equations for numerical models (McDougall et al., 2003), and for the clarification of some misleading explanations for the adiabatic lapse rate, eq. (8), given elsewhere (McDougall and Feistel, 2003). Combined with a simple Gibbs function of ice, see Section 3.3, various properties of sea ice such as freezing point, melting heat or brine salinity could, for the first time, be calculated perfectly consistently

with all other experimental data on seawater and ice involved (Feistel and Hagen, 1998).

Seawater properties are strongly related to those of the pure-water solvent, as given by the Gibbs function $g^{W}$ in eq. (14), and indirectly also by seawater properties measured with instruments that had in advance been calibrated using certain older, possibly inconsistent, pure-water equations. The extremely accurate Helmholtz function for pure water published by Wagner

and Pruß (2002), the so-called IAPWS-95 equation, see Section 3.2, offered the chance for a fundamental update of the pure-water part of the 1995 Gibbs function of seawater. Moreover, it also suggested consistency corrections of the underlying experimental seawater data to be carried out with respect to their respective, obsolete pure-water references, as far as these dependencies could be inferred from the related publications. The resulting new Gibbs function for seawater (Feistel, 2003)



was still expressed in terms of Practical Salinity. For the pure-water part, a Gibbs function had been fitted to IAPWS-95 over the oceanographic ranges of temperature and pressure. Updated ocean-model algorithms were derived from the 2003 equation by Jackett et al. (2006).

At that time, the various substantial advantages of the 2003 Gibbs function over the still valid EOS-80 standard, together with sophisticated new oceanographic tools such as Conservative Temperature, unavailable from EOS-80, sparked the idea of developing a new international seawater standard. In order to investigate this question, at the 2005 SCOR/IAPSO meeting at Cairns, Australia, "IAPSO had introduced a proposal for a new working group, and that was approved by SCOR as Working Group WG127 on Thermodynamics and Equation of State of Seawater. This work is expected to provide important
input into modeling of the global ocean and, ultimately, climate change modeling" (IAPSO, 2005; SCOR, 2005). On the other hand, the Gibbs function of seawater was also presented at the 2005 annual IAPWS meeting at Santorini, Greece. The idea of developing an IAPWS equation for seawater was welcomed there, and resulted in establishing an IAPWS Task Group on Seawater, and in sending a letter to IAPSO expressing the IAPWS interest in working jointly on this problem (IAPWS, 2005).

Chaired by Trevor McDougall, WG127 held its inaugural meeting in 2006 at Warnemünde, Germany, see Appendix B. The WG confirmed the desirability of cooperating with IAPWS on a jointly agreed seawater standard defined by a Gibbs function, suitable for both oceanographic and industrial applications, such as seawater desalination. IAPWS-95 was adopted by WG127 as the pure-water reference for the future thermodynamics of seawater. Detailed numerical checks of the 2003
Gibbs function were initiated. For the standard-ocean reference state ($S_P = 35$, $T = 273.15$ K, $p = 101\,325$ Pa), the WG specified entropy and enthalpy of seawater to vanish. The WG recommended the inclusion of additional available data at normal pressure (heat capacities, dilution heats, freezing points) for an extended Gibbs function.

The WG had an extensive discussion of Absolute Salinity, $S_A$, see Section 3.4. If the thermodynamic functions could be
defined in terms of $S_A$ instead of Practical Salinity $S_P$, this would have the advantage that density calculated from the thermodynamic formulation would be a better fit to actual seawater density because it would reflect compositional changes. If defined by a fixed conversion factor for a reference composition, $S_A$ is as accurate as $S_P$ and fully compatible with present measuring techniques. $S_A$ for a reference composition has an exact relation to traditional chlorinity. All physical, chemical and oceanographic, theoretical as well as numerical, models do actually rely on $S_A$ rather than $S_P$. Outside oceanography, $S_A$
is the only way the scientific community recognises salinity. Salinity in technical/industrial applications (IAPWS) is traditionally understood as Absolute Salinity. The WG gave serious consideration to the idea that the fundamental description should change from Practical Salinity to Absolute Salinity, but postponed decisions on this issue for its wide-range implications (McDougall et al., 2008). However, already in September 2006 some WG members started intense email





activities around the clock and week, across complementary time zones around the globe, aiming at the specification of a best estimate for the chemical composition of Standard Seawater.

In 2007, at the second WG127 meeting held in Reggio, Calabria, see Appendix B, draft articles had already been prepared regarding the Reference Composition (Millero et al., 2008), and an updated Gibbs function (Feistel, 2008a) based on that composition and expressed in terms of Absolute Salinity, $S_A$, which for SSW equals the Reference-Composition Salinity, $S_R$, see eq. (11). While the pressure-independent part (responsible for freezing point, dilution heat, etc.) of the 2008 Gibbs function was significantly extended in its $T$-$S_A$ validity range, the pressure-dependent part (responsible for density and its derivatives) remained identical with that of 2003. WG127 endorsed the use of the IAPWS 2006 Release on an equation of state for H₂O Ice Ih (IAPWS R10-06, 2009). Later, the new salinity definition and the 2008 Gibbs function were also presented at the 2007 IAPWS meeting at Lucerne, Switzerland. On this basis, a "Release on the IAPWS Formulation for the Thermodynamic Properties of Seawater" had been drafted for evaluation, with approval anticipated for 2008 (IAPWS, 2007).

While salinity rarely exceeds 40 g kg⁻¹ in the global ocean, more concentrated brines are found, for example, in the lagoon-like Australian Shark Bay (up to 70 g kg⁻¹, Logan and Cebulsk, 1970), and in particular in brine pockets of polar sea ice at low temperatures. Due to the latent-heat contributions of melting or freezing pockets, the heat capacity of sea ice is significantly larger than that of either pure ice or seawater. Such high-salinity effects could properly be estimated by so-called Pitzer equations (Feistel and Marion, 2007), and so there was a reasonable interest in implementing this ability also in TEOS-10. The new Reference-Salinity Scale supported this extension beyond the validity range of the previous Practical Salinity, see Section 3.4, and at the same time raised the question of solubilities of the dissolved salts (Marion et al., 2009). It turned out that salinities up to 110 g kg⁻¹ of Antarctic sea ice are well covered by the 2008 Gibbs function (Feistel et al, 2010a), see Fig. 4.

At the International Conference on the Properties of Water and Steam 2008 in Berlin, the 2008 Gibbs function was adopted as an IAPWS standard (IAPWS R13-08, 2008), in combination with the IAPWS-95 equation which provides the pure-water part $g^W(T, p)$ of eq. (14) in implicit form. On the basis of mainly these two documents, TEOS-10 was endorsed by IOC/UNESCO in 2009 at Paris as a new international seawater standard "to replace EOS-80 and thus updating this valuable, but no longer state-of-the-art, 30-year-old UNESCO standard" (IOC, 2009, page 5 therein; Wright et al., 2010b; IUGG, 2011; Valladares et al., 2011a,b).

For industrial applications, two modifications of the 2008 Gibbs function for seawater were developed later. First, the validity of a new density equation (Feistel, 2010) at atmospheric pressure could be extended to temperatures up to 90 °C and Absolute Salinities up to 70 g kg⁻¹ using new measurements of Millero and Huang (2009). Second, for an industrial seawater





standard the "scientific" Helmholtz function forming the pure-water part was replaced by an "industrial" IAPWS Gibbs function for liquid water (Wagner and Kretzschmar, 2008; Kretzschmar et al., 2015; IAPWS AN5-13, 2016).

The 2008 Gibbs function for seawater relies mainly on older experimental data of the 1960s and 1970s. Meanwhile, various new measurements of seawater properties, also under conditions outside the validity of TEOS-10, have been published, such as of density (Millero and Huang, 2009; Safarov et al., 2009, 2010, 2012, 2013) and of sound speed (Millero and Huang, 2011; von Rohden et al., 2015, 2016; Lago et al., 2015). Further available data are reviewed by Sharkawy et al. (2010) and Nayar et al. (2016).

**3.6 Helmholtz Function of Humid Air**

The flux of water across the ocean-atmosphere interface belongs to the most important processes of the global climate system, but its estimated contribution to the total heat loss of the ocean varies greatly between 50 % (Emery et al., 2006) and 90 % (Wells, 2012). The thermodynamic driving force for evaporation is the difference between the chemical potentials of water in seawater and in humid air. Usually, this difference is approximated by the relative humidity of air (Kraus and Businger, 1994). This estimate is known to be in error by typically 2 % because of the lowered vapour pressure of seawater compared to that of pure water. To get an idea of the relevance of this error, note that a small change of the global latent heat flux by 1 %, or about 1 W m$^{-2}$, would by a factor of 200 exceed the flux responsible for the currently observed greenhouse warming of the atmosphere, about 5 mW m$^{-2}$ (Lovell-Smith et al., 2016; Feistel, 2015, 2017). The estimated energy imbalance of 0.4 – 0.8 W m$^{-2}$ of the warming ocean (Cheng et al., 2016) is also within this error range. However, on the other hand, routine meteorological measurement of relative humidity has a typical uncertainty between 1 and 5 %rh (Lovell-Smith et al., 2016), thus being insufficient to recognise those crucial heat flux anomalies.

In order to consistently include in TEOS-10 also thermodynamic properties of the air-sea interface, such as latent heat of evaporation or relative humidity at equilibrium with seawater, the first intention was to simply adopt a thermodynamic potential for humid air from the scientific literature. Unfortunately, as it turned out, such a function had never been developed yet by the atmospheric and humidity communities. Similar to the situation with seawater, only certain collections of separate empirical property equations of unclear mutual consistency were available (Goff and Gratch, 1945; Sonntag, 1966; Linke and Baur, 1970; Gill, 1982; Gatley, 2005; Murphy and Koop, 2005; WMO, 2008). So it was necessary to construct a new TEOS-10 Helmholtz function for humid air, $f^{AV}$, from a minimum number of available, mutually independent but internally consistent, empirical *de-facto* standard functions,

$$f^{AV}(A, T, \rho) = Af^{A}(T, A\rho) + (1 - A)f^{V}(T, (1 - A)\rho) + A(1 - A)f^{mix}(A, T, \rho). \qquad (15)$$

Here, $A$ is the mass fraction of dry air in humid air, $\rho$ is the mass density of humid air, $f^{A}$ is the Helmholtz function of dry air of Lemmon et al. (2010), $f^{V}$ is the IAPWS-95 Helmholtz function of water vapour, see Section 3.2, and $f^{mix}$ consists of second (Harvey and Huang, 2007) and third (Hyland and Wexler, 1983) cross-virial coefficients for air-water interaction. Eq.

(15) and some resulting properties, such as for the latent heat of seawater, see Fig. 5, or for the entropy of clouds, see Fig. 6, were first discussed at the 2009 meeting of WG127 at Arnhem, The Netherlands, see Appendix B. In 2010, the Helmholtz function (15) for humid air was eventually adopted as an official IAPWS formulation at Niagara Falls, Canada (IAPWS G8-10, 2010; Feistel et al., 2010b), and presented at the Portorož symposium on temperature and humidity metrology (Feistel, 2012).

In their common ranges of validity, the IAPWS equation of humid air is consistent with the metrological high-accuracy CIPM-2007 equation (Picard et al., 2008) for the density of humid air. The latter paper outlines previous problems with the molar mass of dry air, which differs slightly also between the equations of Feistel et al. (2010b) and IAPWS G8-10 (2010).

Unlike the original definition of Lemmon et al. (2010), the TEOS-10 equation for dry air specifies zero entropy and zero enthalpy of dry air at the standard ocean surface, 0 °C and 101 325 Pa, as reference-state conditions. Ideal-gas approximations and their related adjustable constants, consistent with TEOS-10, are reported in Feistel et al. (2010b). For most atmospheric applications, small corrections to the ideal-gas equations in the form of virial coefficients are sufficiently accurate. For this purpose, from eq. (15) a numerically more convenient virial Gibbs function of humid air can be derived

(Feistel et al., 2015; IAPWS G11-15, 2015). A recent review of thermodynamic equations for humid-air properties is given by Herrmann et al. (2017).

## 4 Extracting Properties from TEOS-10

The four fundamental thermodynamic potentials of TEOS-10, see Fig. 7, as described in Section 3, possess axiomatic properties of consistency, independence and completeness. An axiomatic approach to formally defining and representing all

thermodynamic properties with respect to a minimum common set of basic functions may avoid confusion, may more easily permit identification and quantification of differences between seemingly equivalent quantities such as various alternative definitions of relative humidity, and may establish solid thermodynamic links between quantities that were originally introduced separately and independently (Feistel et al., 2016), such as correlation equations for the heat capacity and for the sublimation pressure of ice, see Section 3.3.

Because of *consistency*, it is impossible to derive from TEOS-10 different results for one and the same thermodynamic quantity. Maxwell's cross-relations are always identically fulfilled, rather than just approximately. In contrast, previous collections of separate property equations are not necessarily consistent. As an example, from the EOS-80 collection (Unesco, 1983) one can compute the heat capacity directly by one given equation, but also indirectly by given equations for

density and sound speed, and those two results differ, especially near the temperature of maximum density.



*Independence* expresses the fact that no part of the four fundamental potentials of TEOS-10 may be derived from other parts. In particular, if a more accurate update is available for any of those parts, independence permits replacing an obsolete part without adjustments to be made for the other parts. In contrast, previous collections of separate property equations are not necessarily independent. As an example, the EOS-80 equation for the pressure dependence of the heat capacity is related by

a Maxwell relation to the temperature dependence of the density. It is therefore not advisable to update one equation of such a collection without carefully considering any possible side effects this may have on the other equations.

*Completeness* of thermodynamic potentials was discovered by Gibbs, see the title-page quotation. As soon as for a given equilibrium system a thermodynamic potential is available, no matter which one, see Section 2, all its thermodynamic

properties can be derived by merely mathematical manipulations. The system may even be a chemically reacting one or a multi-phase composite. In contrast, previous collections of separate property equations are not necessarily complete. As an example, the EOS-80 collection does not provide entropy, enthalpy or chemical potentials of seawater.

Note, however, that there are some minor, usually irrelevant, exceptions to these general rules in TEOS-10. For example, the

thermodynamic potentials are not perfectly consistent as they use slightly different values for the molar gas constant that were the international reference values at the time the particular equation was developed. The thermodynamic potentials are not entirely independent as they commonly depend, for example, on the definition of the temperature scale, see Section 3.1, and on the definitions of reference-state conditions, see Section 2. Completeness does not apply to, say, the sound speed in ice, which is typically anisotropic in crystalline solid states while the TEOS-10 equation describes ice as an isotropic

substance.

Mathematical formulas for the computation of numerous quantities are derived and explained in the TEOS-10 Manual (IOC et al., 2010) and in several IAPWS documents supporting TEOS-10 (IAPWS AN6-16, 2016). Large subsets of those quantities are implemented in two open-source libraries, the Sea-Ice-Air (SIA) library (Feistel at al., 2010a; Wright et al.,

2010a), and the Gibbs-Seawater (GSW) library (McDougall and Barker, 2011; McDougall et al., 2012; Roquet et al., 2015). Source code in several programming languages is freely available from the TEOS-10 web site, www.teos-10.org.

In the SIA library, thermodynamic potentials are numerically available for single-phase and multi-phase systems such as those displayed in Fig. 7, together with various properties such as heat capacities, densities or entropies of those systems. As

a first step, for a convenient calculation of properties at given temperature, pressure and composition, Gibbs functions are implemented for seawater, $g^{SW}(S_A, T, p)$, and humid air, $g^{AV}(A, T, p)$, together with 9 additional functions each, for any of their 1st and 2nd partial derivatives. For this purpose, equations for the pressure as a function of the density, $p = \rho^2(\partial f/\partial \rho)$, available from the Helmholtz functions, see Section 2, must be inverted iteratively to find the density as a function of the given pressure. The specific entropy, $\eta$, of a sample with given in-situ temperature $T$ at a pressure $p$ is then available from the





related Gibbs function by evaluating $\eta = -\partial g/\partial T$. As a second step, enthalpies $h^{\mathrm{SW}}(S_A, \eta, p)$ and $h^{\mathrm{AV}}(A, \eta, p)$ of seawater and humid air, respectively, are implemented by executing another numerical iteration procedure. From those functions, for a given reference pressure $p_{\mathrm{ref}}$, potential enthalpy, $h_\theta = h(\eta, p_{\mathrm{ref}})$, potential temperature, $\theta = \partial h(\eta, p_{\mathrm{ref}})/\partial\eta$, and potential density, $\rho_\theta^{-1} = \partial h(\eta, p_{\mathrm{ref}})/\partial p$, can be computed. Note that all input and output quantities of the SIA library are expressed

in basic SI units, such as temperatures in K (rather than °C), pressures in Pa (rather than dbar relative to the surface pressure), and salinity or dry-air fraction in kg kg$^{-1}$ (rather than in psu or g kg$^{-1}$ or %). This strict convention avoids the need for any unit conversions along with the various mathematical manipulations performed internally. Such conversions are error-prone; to see this, imagine to modify the sound speed formula (7) for the case that the pressure is measured in decibars rather than pascals.

Similar to the potential functions and properties of single-phase systems, the SIA library implements also Gibbs functions and enthalpies of composite systems, such as sea ice, consisting of seawater and ice, or clouds, consisting of liquid water and humid air, at their mutual thermodynamic phase equilibria. The SIA library is organised in the form of several separate modules. It is possible to select an axiomatic subset of such modules as a sub-library, such as one just for ice, or one for

humid air, as discussed in the digital supplement of Feistel et al. (2016). It is also possible to combine SIA modules with additional user-defined modules for, say, specific functions that do not yet belong to the standard library set. For example, such add-ons may be developed for including the effects of dissolved air on water properties which are currently neglected in TEOS-10.

For frequent calls within ocean models, the stacked iterations of the SIA library are too slow, and the variables and their units are inconvenient. For these reasons, the GSW library implements fast and tailored equations in terms of appropriate input parameters, accepting oceanographically familiar units, such as pressure in decibars, Absolute Salinity in grams per kilogram, and Conservative Temperature as potential enthalpy expressed in °C. The price for this advantage is the introduction of new empirical equations with additional regression coefficients obtained by fitting with respect to data

calculated from the SIA library equations. Potential future updates of any fundamental TEOS-10 equation, see Fig. 7, will automatically propagate into the various derived SIA functions, but will require updated sets of coefficients of the GSW library. Detailed descriptions of the oceanographic quantities and their use in the context of TEOS-10 are available from IOC et al. (2010), McDougall and Barker (2011), McDougall et al. (2013) and Roquet et al. (2015).

## 5 Problems Beyond TEOS-10

TEOS-10 was developed as a state-of-the-art thermodynamic framework supporting geosciences in their tasks of observing, understanding, modeling and predicting the global climate system and its long-term variations. Successful cooperation between climate-related research groups anywhere on the planet, extending over several human generations, demands mutual



comparability of their respective individual theoretical and measurement results. This indispensable requirement appears to be self-evident, but unfortunately it is by no means always fulfilled yet, and in certain cases it even poses a serious challenge still to be addressed in the future. All relevant quantities should be uniformly defined, preferably by commonly agreed international standards or recommendations. All measurement results of the same kind of quantity should be metrologically

traceable to the same reference, preferably specified as an international standard of highest temporal stability. The actual quantities of interest, such as seawater salinity, should be related to their associated measurands, such as temperature and conductivity, by standard equations of certified estimated uncertainty. Related pending problems were presented and discussed at the 2010 WMO-BIPM workshop on Measurement Challenges for Global Observation Systems for Climate Change Monitoring in Geneva, Switzerland (WMO, 2010).

According to the rules for SCOR/IAPSO Working Groups, WG127 was disbanded in 2011 (Pawlowicz et al., 2012). During the development of TEOS-10, relevant problems had become visible regarding the definitions of seawater salinity, seawater pH, and relative humidity of moist air. In order to address these problems along with maintaining TEOS-10, a standing Joint Committee on the Properties of Seawater, JCS, was commonly established by SCOR, IAPSO and IAPWS in 2012 (IAPWS,

2012). At the BIPM at Sèvres in 2011 and 2012, related meetings on a potential future cooperation took place, and subsequently a joint workshop of JCS was held with representatives of the BIPM at the 2013 International Conference on the Properties of Water and Steam in Greenwich, UK (Feistel, 2013; IAPWS, 2013; Hellmuth et al., 2014; Pawlowicz et al., 2014; Feistel et al., 2016). Recent progress and upcoming tasks are planned to be on the agenda of the 2018 International Conference on the Properties of Water and Steam in Prague, Czech Republic.

**5.1 Seawater Salinity**

During the development of TEOS-10, much attention was paid to the question of how salinity should be defined in oceanography (McDougall et al., 2008; Millero et al., 2008; Seitz et al., 2011; Wright et al., 2011; Pawlowicz et al., 2016). In current practice, salinity measurement is performed by electronic sensors for pressure, temperature and conductivity. Those sensors usually consist of some arrangement of electronic devices, such as a platinum resistor, and typically return

signals in the form of an electric voltage, a current, or a frequency. In a calibration lab, the sensor is immersed in seawater of well-defined conditions, and its output signal values are associated with the particular temperatures, pressures or salinities of the bath. To establish those well-defined conditions, the lab needs to measure the bath temperature and to use seawater samples of certified salinities. The lab's thermometers are regularly calibrated against temperature standards realised in national metrological institutes, which in turn implement the rules specified for the ITS-90 definition of the kelvin within the

International System of Units (SI). Certified Standard Seawater samples are commercially produced and distributed by the IAPSO Standard Seawater Service operated by OSIL, Ocean Scientific International Ltd., a company located near Portsmouth, UK. At OSIL, seawater samples collected from the North Atlantic are purified, diluted and compared against the conductivity of a potassium chloride (KCl) solution as specified by 1978 Practical Salinity Scale. These chains of



calibration procedures establish the so-called *metrological traceability* of measurement results (de Bievre and Günzler, 2005; Seitz et al., 2011; VIM3, 2012). Traceability to the same primary reference is a necessary condition for the comparability of measured values from different devices, at different locations or times. While temperature measurement is traceable to the SI, salinity is traceable to manufactured, possibly varying or aging, artefacts in the form of certain KCl solutions or certified
SSW batches (Seitz et al., 2011).

For long-term series such as required for climate research, the temporal stability of the primary references is of ultimate relevance. It must be granted that tiny trends in observed values constitute real changes rather than just spurious effects, caused by drifts of measurement devices or standards. In order to achieve the highest metrological stability available,
traceability to the SI is the preferred choice. However, traceability to the SI is of sufficient advantage only if the related measurement uncertainty is small enough. For example, absolute conductivity measurements are traceable to the SI, but their uncertainty is about one order of magnitude too large compared to the PSS-78 standard, and defining salinity by a specified conductivity number rather than by a mass fraction of a reference KCl solution, would in practice let small but significant observed salinity trends disappear in point clouds of noisy scatter. Based on such arguments, after many long and vivid
discussions, the 2008 meeting of WG127 at Goetz near Berlin, see Appendix B, came to the conclusion that currently, seawater density is the only promising candidate for SI-traceable salinity measurements in the oceanographic practice (Seitz et al., 2011; Wright et al., 2011; Pawlowicz et al., 2016).

TEOS-10 has already established density measurement as a secondary method for the determination of Absolute Salinity in
cases of expected seawater composition anomalies (Millero et al., 2008; Feistel at al., 2010c), see Section 3.4. The concept recently developed by JCS (Pawlowicz et al., 2016) aims at a specification of certified Standard Seawater samples not only by their Practical Salinities as presently, but in the future also in parallel by their densities, measured in a way that is traceable to the SI. This approach would leave the current oceanographic practice unaffected but could grant the requisite long-term stability of the SSW standard, for the very first time after SSW was introduced by Knudsen more than a century
ago (Knudsen, 1903; Wallace, 1974; Culkin and Smed, 1979; Burchard et al., 2018). Similar to the PSS-78 standard, which specifies a standard equation for the conversion of measured triples of pressure, temperature and conductivity ratio to Practical Salinity, the related standard equation for the density-salinity conversion would be the TEOS-10 equation of state, as already implemented in the SIA and GSW software libraries. While this concept appears physically and metrologically sound, there is a large number of detailed technical, metrological, logistical and financial questions that need to be addressed
before a new international, density-based salinity standard may be introduced in the future (Pawlowicz et al., 2016; Schmidt et al., 2016, 2018).





## 5.2 Seawater pH

As a marine chemistry quantity, the pH of seawater does not belong to the thermodynamic properties available from the equations of TEOS-10, but connections between them were on the agenda of WG127 already in 2007 at Reggio, see Appendix B, and at later meetings. As an urgent problem beyond TEOS-10, seawater pH belongs to the climatological key

parameters depicted as metrological challenges by JCS, the successor of WG127 (Feistel et al., 2016; Dickson et al., 2016). For suspectedly disastrous effects on marine ecosystems, the acidification of the oceans, quantified in terms of seawater pH values, has become a severe public, ecological and political concern (Feely et al., 2004; Le Quéré et al., 2015), far beyond mere curiosity of a few scientists. However, already the inventor of the pH value, Sören Peter Lauritz Sörensen, had lamented the ambiguous measurement methods in use for seawater pH. "So wäre es zu wünschen, daß man zukünftig bei der

Bestimmung … immer von denselben Voraussetzungen ausgeht, oder jedenfalls daß die Grundlage des angewandten Verfahrens scharf pointiert wird" (Sörensen and Palitzsch, 1910, p. 415 therein) [English translation: „It is desirable that future estimates … be always based on the same assumptions, or at least that the method applied be precisely described"]. Unfortunately, over the past century this situation has improved only insignificantly. Different seawater pH scales are in practical use, often denoted by simply "pH", which may deviate from one another stronger than the expected variations to be

resolved (Marion et al., 2011; Spitzer et al. 2011; Brewer 2013; Dickson et al., 2016). Indeed, still "the field of pH scales and the study of proton-transfer reactions in sea water is one of the more confused areas of marine chemistry" (Dickson, 1984).

Originally, Sörensen (1909) had defined the pH value of an aqueous solution as the negative common logarithm of the hydrogen-ion concentration expressed in moles per litre. To better account for effects of ionic interactions in the solution, in

this definition equation the concentration was later replaced by the hydrogen-ion activity, specified in a way that both definitions coincide asymptotically in the ideal-solution approximation (Sørensen and Linderstrøm-Lang, 1924). However, even without application to seawater in particular, these definitions involve two severe physico-chemical and metrological difficulties.

The first problem is that physically any distinction between a free (dissociated) ion and a bound (associated) ion is subject to arbitrary convention (Bjerrum, 1926; Falkenhagen et al., 1971; Ebeling and Grigo, 1982; Justice, 1991). Consequently, direct measurements of ion concentrations of incompletely dissociated solutions are impossible unless the underlying convention is somehow embodied in the measurement procedure. What in fact can be measured are certain properties correlated with ion concentrations, such as the colour of an indicator dye by photometric pH methods. For obtaining the quantity of interest (pH)

from the quantity actually measured (colour), a calibration relation is required which implicitly or explicitly implements the particular ion-association convention, similar to the ticks on a mercury thermometer that implement the definition of the Celsius scale in terms of length units. However, the requisite pH convention is pending yet.





The second problem is that single-ion activities cannot be measured either (Bjerrum, 1919; Guggenheim, 1949), nor can they unambiguously be inferred from mean chemical potentials of electrically neutral combinations of ions. To overcome this problem, auxiliary assumptions are sometimes applied, such as equating the activities of cations and anions of a particular solute, as had been suggested for KCl by Lewis and Randall (1923). Such arbitrary practical "conventions" may reasonably

be applied as long as they do not conflict with experimental evidence. On the other hand, in contrast to empirical thermodynamics, in statistical thermodynamics of electrolytes the Debye-Hückel limiting law predicts the single-ion activity to be a well-defined function of the ionic strength of very dilute electrolytes (Falkenhagen et al., 1971; Prausnitz et al., 1999). Theoretical relations of this kind between activities and other measurable quantities, such as equations for single-ion activities derived from Pitzer equations, are at odds with the putative validity of artificially constructed conventions.

Beyond the limiting law, analytical expressions for single-ion activities or related analytical expressions are only approximately available for dilute solutions (below 1 mol l$^{-1}$) in the theoretical framework of statistical thermodynamics of electrolytes (Wiechert et al., 1978; Ebeling and Scherwinski, 1983). At ion concentrations such as 1.2 mol kg$^{-1}$ typically encountered in the ocean (Feistel and Marion, 2007), convenient simplifying theoretical models such as hard spherical ions

immersed in an unstructured background solvent become increasingly inappropriate. The real microscopic interactions of electrolytes are not precisely known and can only approximately be represented mathematically. A workable practical approach to this problem may be the use of so-called Pitzer equations. They approximate single-ion activities as series expansions with respect to the ion concentrations and adjust the unknown empirical coefficients to other, measurable properties (Nesbitt, 1980; Marion and Grant, 1994; Prausnitz et al., 1999; Marion and Kargel, 2008; Marion et al., 2011).

Pitzer equations for seawater ions successfully describe colligative properties while other thermodynamic properties such as sound speed may not yet be represented as accurate as by TEOS-10 (Feistel and Marion, 2007; Feistel, 2008a; Sharp et al., 2015). Based on the Reference Composition model of TEOS-10 (Millero et al., 2008), studies are underway aiming at a definition of seawater pH in terms of Pitzer equations as functions of SI-traceable measurands (Waters and Millero, 2013; Dickson et al., 2016; Turner et al., 2016; Camoes et al., 2016).

**1.2 Relative Humidity**

In TEOS-10, equations for two different definitions of relative humidity (RH) are provided, for the "WMO definition" and the "CCT definition" (IOC et al., 2010; Feistel et al., 2010a,b; IOC et al., 2010). These choices made for both names turned out to be misleading, unfortunately. The so-called "WMO definition" was taken over from recent textbooks on atmospheric sciences (Gill, 1982; Rogers and Yau, 1989; Pruppacher and Klett, 1997; Jacobson, 2005; Pierrehumbert, 2010), however,

the definition reported in those books incorrectly retains an obsolete definition of the OMI (1951) which was superseded by the actual WMO definition already in 1954. What is called the "CCT definition" in TEOS-10, on the other hand, is in fact the current WMO definition (WMO, 2008), while the Consultative Committee on Thermometry (CCT) of the BIPM has not recommended yet any RH definition by now. Subsequently, a closer inspection showed that this regrettable naming



confusion in TEOS-10 reveals only the "tip of the iceberg" (Lovell-Smith et al., 2016). Several mutually inconsistent definitions of relative humidity prevail in different branches of meteorology, climatology, physical chemistry, food engineering or air conditioning, all of them typically denoted by just "RH" without additional specification.

The global ocean surface presents a vast open window to the very heart of the dynamics governing the climate system. Properties and processes may be observed there by research vessels, automatic buoys and remotely sensing satellites over long times with exceptional spatial coverage. On the other hand, clouds, wind, waves, corrosive and depositing sea salt, floating icebergs, plastic garbage or oil spills, commercial ship traffic and fishery, sensor fouling by biological processes, political, economic and military interests of riparian states, and large distances and high costs for device maintenance may
pose severe difficulties for systematic scientific investigations of the air-sea interface at the open ocean. While TEOS-10 cannot overcome such practical obstacles, it has been developed with the intention to provide a consistent thermodynamic framework for studies of the transfer of energy and water between ocean and atmosphere as key players of the terrestrial climate.

Water in the troposphere has a mean residence time of 8-10 days until it precipitates as rain or snow. About 80 % to 90 % of that water is replaced by evaporation from the oceans (Reid and Valdés, 2011). Evaporation is driven by the difference between the chemical potentials of water in humid air and in seawater or ice. Per mass of water, these chemical potentials are available from TEOS-10; for humid air from the Helmholtz function (15) by the relation (IAPWS G8-10, 2010),

$$\mu_W^{AV} = f^{AV} + \rho \frac{\partial f^{AV}}{\partial \rho} - A \frac{\partial f^{AV}}{\partial A}, \tag{16}$$

for seawater (and in the limit $S_A = 0$, also for pure liquid water) from the Gibbs function (14) by the relation (IAPWS R13-08, 2008),

$$\mu_W^{SW} = g^{SW} - S_A \frac{\partial g^{SW}}{\partial S_A}, \tag{17}$$

and for ice from its Gibbs function, $g^{Ih}$, by the relation (IAPWS R10-06, 2009),

$$\mu_W^{Ih} = g^{Ih}. \tag{18}$$

Chemical potentials per mole of water, $\mu$, may similarly be expressed by so-called *fugacities*, $f$, (symbol not to be confused here with the Helmholtz functions, $f$, considered earlier), according to the relation (Guggenheim, 1949; Prausnitz et al., 1999)

$$\mu = \mu^{id} + RT ln \frac{f}{xp}. \tag{19}$$

Here, $x$ is the mole fraction of water in the substance considered, $\mu^{id}$ is the ideal-gas chemical potential of water vapour with
the same number of moles, and $R$ is the molar gas constant. It is obvious from eq. (19) that the fugacity, $f$, plays the role for a real gas which the partial pressure, $xp$, plays for an ideal gas (Lewis, 1901). Regardless of the wind conditions, evaporation from the ocean ceases when the fugacity of water in seawater equals the fugacity of water in humid air. This fact makes hot





humid tropics so unpleasant to humans because sweating provides no cooling there. A statement similar to that for the fugacities may not be made for partial pressures, however, as neither a fictitious "partial pressure of water in seawater" is a well-defined quantity, nor does the partial pressure of water in humid air state any thermodynamic criterion for phase equilibria. For these physical reasons, TEOS-10 suggests a replacement of the usual definition of relative humidity in terms

of partial pressures by a new definition in terms of fugacities (IOC et al., 2010; Feistel et al. 2010b, 2015, 2016; Feistel, 2012; IAPWS G11-15, 2015; Feistel and Lovell-Smith, 2017). In fact, the small real-gas effects of the atmosphere are of similar magnitude as the expected changes of relative humidity associated with greenhouse effects (Hellmuth et al., 2018).

## 6 Summary

TEOS-10 is the present international standard for thermodynamic properties of water, ice, seawater and humid air,

recommended for geosciences by leading international organisations such as UNESCO/IOC and IUGG. TEOS-10 incorporates an extended manifold of different experimental data for those substances, collected from publications over many decades, in an unprecedentedly compact, consistent, comprehensive and accurate way, based on an axiomatic approach that offers various advantages over otherwise used, relatively arbitrary collections of empirical property equations with often unclear mutual consistency. However, certain important problems have remained to be addressed even after the introduction

of TEOS-10. In particular, the climatological key parameters ocean salinity, seawater pH and atmospheric relative humidity still pose metrological challenges that are currently investigated by the standing SCOR/IAPSO/IAPWS Joint Committee on the Properties of Seawater, JCS, in cooperation with other international organisations.

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

## Appendix A: The Jacobi Method

Consider $N$ functions, $y_1$, $y_2$, ... $y_N$, depending on $N$ independent variables, $x_1$, $x_2$, ... $x_N$. The matrix $A = \{a_{ij}\}$ consisting of their pairwise partial derivatives, $a_{ij} = \partial y_i / \partial x_j$, as its elements is regarded as the *Jacobian matrix* of this set of functions. The determinant $J = |A| = \det\{a_{ij}\}$ of this matrix is commonly referred to as the *functional determinant*, or the *Jacobian*





(Bronstein and Semendjajew, 1979; Kaplan, 1984; Gradshteyn and Ryshik, 2000). It is instructive to write $J$ in the form of a fraction,

$$J = \det\left\{\frac{\partial y_i}{\partial x_j}\right\} = \frac{\partial(y_1, y_2, \ldots y_N)}{\partial(x_1, x_2, \ldots x_N)}. \tag{A.1}$$

For $N = 1$, the Jacobian agrees with the partial derivative,

$$J = \frac{\partial(y_1)}{\partial(x_1)} = \frac{\partial y_1}{\partial x_1} \quad . \tag{A.2}$$

Determinants of higher dimensions $N$ can be computed by means of the so-called *Laplace expansion* with respect to a selected row or column of the matrix and the related remaining submatrices, the *minors* of dimension $N - 1$. For $N = 2$, the Jacobian reads

$$J = \frac{\partial(y_1, y_2)}{\partial(x_1, x_2)} = \frac{\partial y_1}{\partial x_1} \times \frac{\partial(y_2)}{\partial(x_2)} - \frac{\partial y_1}{\partial x_2} \times \frac{\partial(y_2)}{\partial(x_1)} = \frac{\partial y_1}{\partial x_1}\frac{\partial y_2}{\partial x_2} - \frac{\partial y_1}{\partial x_2}\frac{\partial y_2}{\partial x_1} \tag{A.3}$$

where the 2-dimensional Jacobian is expressed in terms of 1-dimensional Jacobians.

For $N = 3$, the Laplace expansion takes the form

$$J = \frac{\partial(y_1, y_2, y_3)}{\partial(x_1, x_2, x_3)} = \frac{\partial y_1}{\partial x_1} \times \frac{\partial(y_2, y_3)}{\partial(x_2, x_3)} - \frac{\partial y_1}{\partial x_2} \times \frac{\partial(y_2, y_3)}{\partial(x_1, x_3)} + \frac{\partial y_1}{\partial x_3} \times \frac{\partial(y_2, y_3)}{\partial(x_1, x_2)} \tag{A.4}$$

where the 3-dimensional Jacobian is expanded into products of 1- und 2-dimensional Jacobians. Eqs. (A.3) and (A.4) are the

most frequently used relations required for deriving TEOS-10 functions with 2 or 3 independent variables.

Here are some useful manipulation rules for Jacobians. General properties of determinants imply that the exchange of any two variables switches the sign of the Jacobian, e.g.,

$$\frac{\partial(y_1, \ldots, y_i, \ldots, y_j, \ldots y_N)}{\partial(x_1, x_2, \ldots x_N)} = -\frac{\partial(y_1, \ldots, y_j, \ldots, y_i, \ldots y_N)}{\partial(x_1, x_2, \ldots x_N)} \quad , \tag{A.5}$$

and similarly,

$$\frac{\partial(y_1, \ldots y_N)}{\partial(x_1, \ldots x_i, \ldots x_j, \ldots x_N)} = -\frac{\partial(y_1, \ldots y_N)}{\partial(x_1, \ldots x_j, \ldots x_i, \ldots x_N)}. \tag{A.6}$$

For $N = 2$, this means

$$\frac{\partial(y_1, y_2)}{\partial(x_1, x_2)} = -\frac{\partial(y_2, y_1)}{\partial(x_1, x_2)} = -\frac{\partial(y_1, y_2)}{\partial(x_2, x_1)} = \frac{\partial(y_2, y_1)}{\partial(x_2, x_1)} \tag{A.7}$$

As special cases for $N = 3$, the reversal of variables inverts the sign,

$$\frac{\partial(y_1, y_2, y_3)}{\partial(x_1, x_2, x_3)} = -\frac{\partial(y_3, y_2, y_1)}{\partial(x_1, x_2, x_3)} = -\frac{\partial(y_1, y_2, y_3)}{\partial(x_3, x_2, x_1)} = \frac{\partial(y_3, y_2, y_1)}{\partial(x_3, x_2, x_1)} \tag{A.8}$$

while the rotation of variables preserves the sign,

$$\frac{\partial(y_1, y_2, y_3)}{\partial(x_1, x_2, x_3)} = \frac{\partial(y_3, y_1, y_2)}{\partial(x_1, x_2, x_3)} = \frac{\partial(y_2, y_3, y_1)}{\partial(x_1, x_2, x_3)} = \frac{\partial(y_1, y_2, y_3)}{\partial(x_3, x_1, x_2)} = \frac{\partial(y_1, y_2, y_3)}{\partial(x_2, x_3, x_1)} \tag{A.9}$$

If one (or more) of the functions is an identity, say, $y_N(x_1, \ldots x_N) \equiv x_N$, the Jacobian reduces by one dimension (or more), due to the Laplace expansion,



$$\frac{\partial(y_1, y_2, \dots y_{N-1}, x_N)}{\partial(x_1, x_2, \dots x_{N-1}, x_N)} = \frac{\partial(y_1, y_2, \dots y_{N-1})}{\partial(x_1, x_2, \dots x_{N-1})}. \tag{A.10}$$

In particular, the identical Jacobian equals unity,

$$\frac{\partial(x_1, x_2, \dots x_{N-1}, x_N)}{\partial(x_1, x_2, \dots x_{N-1}, x_N)} = 1. \tag{A.11}$$

The product rule for functional determinants of higher dimensions is a generalisation of the usual chain rule for partial

derivatives. If a set of functions $z$ depends on the variables $y$, which in turn depend on the variables $x$, the relation between

their Jacobians is (Bronstein and Semenjajew, 1979),

$$\frac{\partial(z_1, z_2, \dots z_N)}{\partial(x_1, x_2, \dots x_N)} = \frac{\partial(z_1, z_2, \dots z_N)}{\partial(y_1, y_2, \dots y_N)} \times \frac{\partial(y_1, y_2, \dots y_N)}{\partial(x_1, x_2, \dots x_N)}. \tag{A.12}$$

In particular, if the functions $z$ are chosen identical to $x$, for the inverse functions there follows from (A.12),

$$\frac{\partial(y_1, y_2, \dots y_N)}{\partial(x_1, x_2, \dots x_N)} = \left[ \frac{\partial(x_1, x_2, \dots x_N)}{\partial(y_1, y_2, \dots y_N)} \right]^{-1} \tag{A.13}$$

Developed by Shaw (1935), the Jacobi method is the mathematically most elegant way of transforming the various partial

derivatives of different thermodynamic potentials into one another, exploiting the convenient formal calculus of functional

determinants (Margenau and Murphy, 1943; Landau and Lifschitz, 1966). Namely, if any thermodynamic derivative in two

variables, $(\partial u / \partial x)_y$, is considered, it can formally be written as a Jacobian, eq. (A.10)

$$\left( \frac{\partial u}{\partial x} \right)_y \equiv \frac{\partial(u,y)}{\partial(x,y)}. \tag{A.14}$$

If this Jacobian is to be expressed in specific variables, say $T$ and $p$, the transformation into these independent variables

follows from (A.12) as

$$\left( \frac{\partial u}{\partial x} \right)_y = \frac{\frac{\partial(u,y)}{\partial(T,p)}}{\frac{\partial(x,y)}{\partial(T,p)}}. \tag{A.15}$$

If any of the variables $u$, $x$, $y$ equals $T$ or $p$, the numerator or denominator of (A.15) can be simplified by means of the rules

(A.7) and (A.10). If not, we get the result for (A.15) from (A.3),

$$\left( \frac{\partial u}{\partial x} \right)_y = \frac{\frac{\partial(u,y)}{\partial(T,p)}}{\frac{\partial(x,y)}{\partial(T,p)}} = \frac{\left( \frac{\partial u}{\partial T} \right)_p \left( \frac{\partial y}{\partial p} \right)_T - \left( \frac{\partial y}{\partial T} \right)_p \left( \frac{\partial u}{\partial p} \right)_T}{\left( \frac{\partial x}{\partial T} \right)_p \left( \frac{\partial y}{\partial p} \right)_T - \left( \frac{\partial y}{\partial T} \right)_p \left( \frac{\partial x}{\partial p} \right)_T}. \tag{A.16}$$

As an example, we compute the sound speed, $c$, eq. (7), in terms of derivatives of the Gibbs function $g(T, p)$, and of the

Helmholtz function, $f(T, \rho)$. For the Gibbs function we get from (A.16)

$$c^2 = \left( \frac{\partial p}{\partial \rho} \right)_\eta = \frac{\partial(p, \eta)}{\partial(\rho, \eta)} = \frac{\frac{\partial(p, \eta)}{\partial(T,p)}}{\frac{\partial(\rho, \eta)}{\partial(T,p)}} = \frac{-\left( \frac{\partial \eta}{\partial T} \right)_p}{\left( \frac{\partial \rho}{\partial T} \right)_p \left( \frac{\partial \eta}{\partial p} \right)_T - \left( \frac{\partial \eta}{\partial T} \right)_p \left( \frac{\partial \rho}{\partial p} \right)_T}. \tag{A.17}$$

Substituting in (A.17) the variables $\rho$ and $\eta$, respectively, by means of $g_p = v = 1/\varrho$ and $g_T = -\eta$, where partial derivatives

of $g$ are now conveniently written as subscripts, the final formula for the sound speed expressed in terms of the Gibbs

function reads

$$c = g_p \sqrt{\frac{g_{TT}}{g_{Tp}^2 - g_{TT} g_{pp}}}. \tag{A.18}$$




Alternatively, for the Helmholtz function we get from (A.16) the sound speed formula, replacing in (A.17) the independent variable $p$ by $\rho$,

$$c^2 = \left(\frac{\partial p}{\partial \rho}\right)_\eta = \frac{\partial(p,\eta)}{\partial(\rho,\eta)} = \frac{\frac{\partial(p,\eta)}{\partial(T,\rho)}}{\frac{\partial(\rho,\eta)}{\partial(T,\rho)}} = \frac{\left(\frac{\partial p}{\partial T}\right)_\rho\left(\frac{\partial \eta}{\partial \rho}\right)_T - \left(\frac{\partial \eta}{\partial T}\right)_\rho\left(\frac{\partial p}{\partial \rho}\right)_T}{-\left(\frac{\partial \eta}{\partial T}\right)_\rho}. \tag{A.19}$$

Substituting in (A.19) the variables $p$ and $\eta$, respectively, by means of $\varrho^2 f_\varrho = p$ and $f_T = -\eta$, the final formula for the sound speed expressed in terms of the Helmholtz function reads

$$c = \sqrt{\frac{-\varrho^2 f_{T\varrho}^2}{f_{TT}} + 2\varrho f_\varrho + \varrho^2 f_{\varrho\varrho}}\,. \tag{A.20}$$

For more independent variables, such as the additional mass fractions of sea salt or dry air, the method applies correspondingly. If a derivative $(\partial u/\partial x)_{y,z}$ is to be calculated, we apply the rule (A.10),

$$\left(\frac{\partial u}{\partial x}\right)_{y,z} \equiv \frac{\partial(u,y,z)}{\partial(x,y,z)}\,. \tag{A.21}$$

If this derivative of an arbitrary function $u(x, y, z)$ needs to be expressed in, say, the particular variables $S$, $T$, $p$ of the Gibbs function of seawater, eq. (A.21) needs to be transformed to those variables by the thermodynamic "chain rule"

$$\left(\frac{\partial u}{\partial x}\right)_{y,z} \equiv \frac{\partial(u,y,z)}{\partial(x,y,z)} = \frac{\partial(u,y,z)}{\partial(S,T,p)}\bigg/\frac{\partial(x,y,z)}{\partial(S,T,p)}\,. \tag{A.22}$$

Then, the Jacobians in the numerator and denominator are evaluated by the rules (A.4), (A.8), (A.9) and (A.10), until (A.22) is eventually expressed by the Gibbs function $g$ and its derivatives with respect to $S$, $T$ and $p$.

## Appendix B: Meetings of SCOR/IAPSO WG127

WG127 was formed of a small group of specialists from countries all over the world. After the WG had been established in 2005, the WG Chair, Trevor McDougall, was its first, natural member who decided to invite Rainer Feistel as a second member. Together possible further candidates were then commonly discussed and contacted as the next potential member. So the democratic invitation process went on up to the intended maximum size of the group. Rather than face to face, many of the members knew each other only from publications before. Despite this, the work of WG127 was very constructive and intense from the first day on. Year by year, the membership varied slightly, bringing new expertise into the discussions and decisions. Until the adoption of TEOS-10 in 2009, the participants of the four WG127 meetings may briefly be reported here as an appreciation and gratitude for their substantial contributions and very successful cooperation during those years. It was very sad, however, that two extremely productive and creative friends and colleagues were lost quite untimely: Dan Wright died in 2010, and David Jackett in 2012. In memory, TEOS-10 will always be associated with their names.

*2006 Initial Meeting at Warnemünde, Germany, 2 to 5 May, see Fig. B.1*
*2007 Meeting at Reggio/Calabria, Italy, 7 to 10 May, see Fig. B.2*
*2008 Meeting at Goetz near Berlin, Germany, 3 to 9 September, see Fig. B.3*
*2009 Final Meeting at Arnhem, The Netherlands, 2 to 5 September:*

Participants (alphabetically, no photo available): Rainer Feistel (Germany), Brian A. King (UK), Norge Larson (USA), Trevor J. McDougall (Australia), Richard Pawlowicz (Canada), Daniel G. Wright (Canada).




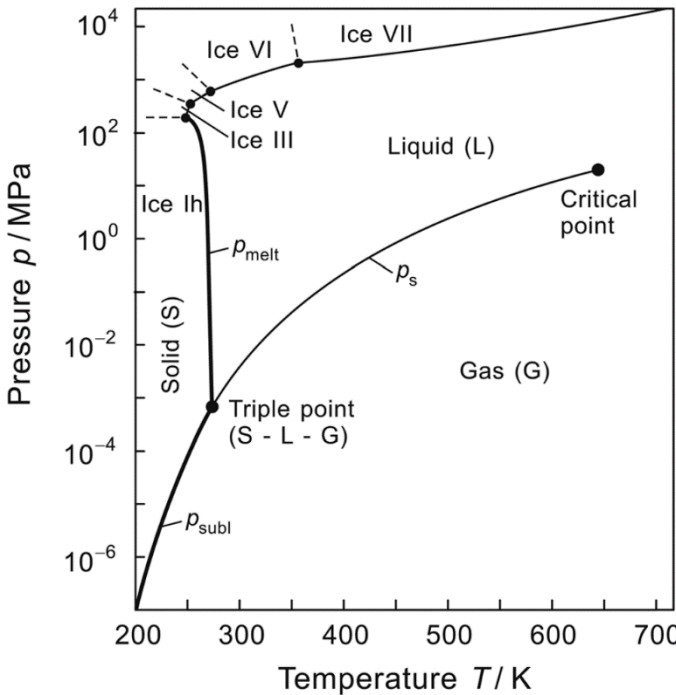

**Figure 1: Temperature-pressure diagram of water (from IAPWS R14-08, 2011, permitted). The curves indicate phase transitions between stable gaseous, liquid and solid states; $p_{melt}$ indicates the melting line of ambient hexagonal ice Ih, $p_{subl}$ its sublimation line, and $p_s$ the saturation line, or boiling line, between liquid and gas. Several additional ice phases are not indicated. Within this diagram, geophysical conditions for ice, liquid water and water vapour cover only small regions (not shown) in some vicinity of the triple point.**



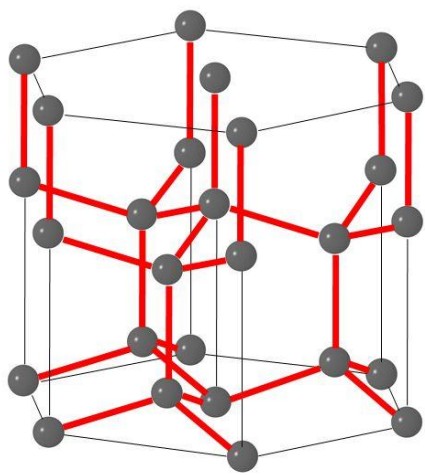

**Figure 2: The hexagonal elementary crystal of ice Ih (Penny, 1948; Schulson, 1999) consists of 27 oxygen (O) atoms (spheres) and 28 hydrogen (H) bonds between them (bars). Of the four H-atoms adjacent to each O-atom, two are placed closer than the other pair, thus retaining the structure of individual $H_2O$ molecules within the crystal lattice. Despite the crystal's anisotropy, thermodynamic ice properties are isotropic to within measurement uncertainty.**





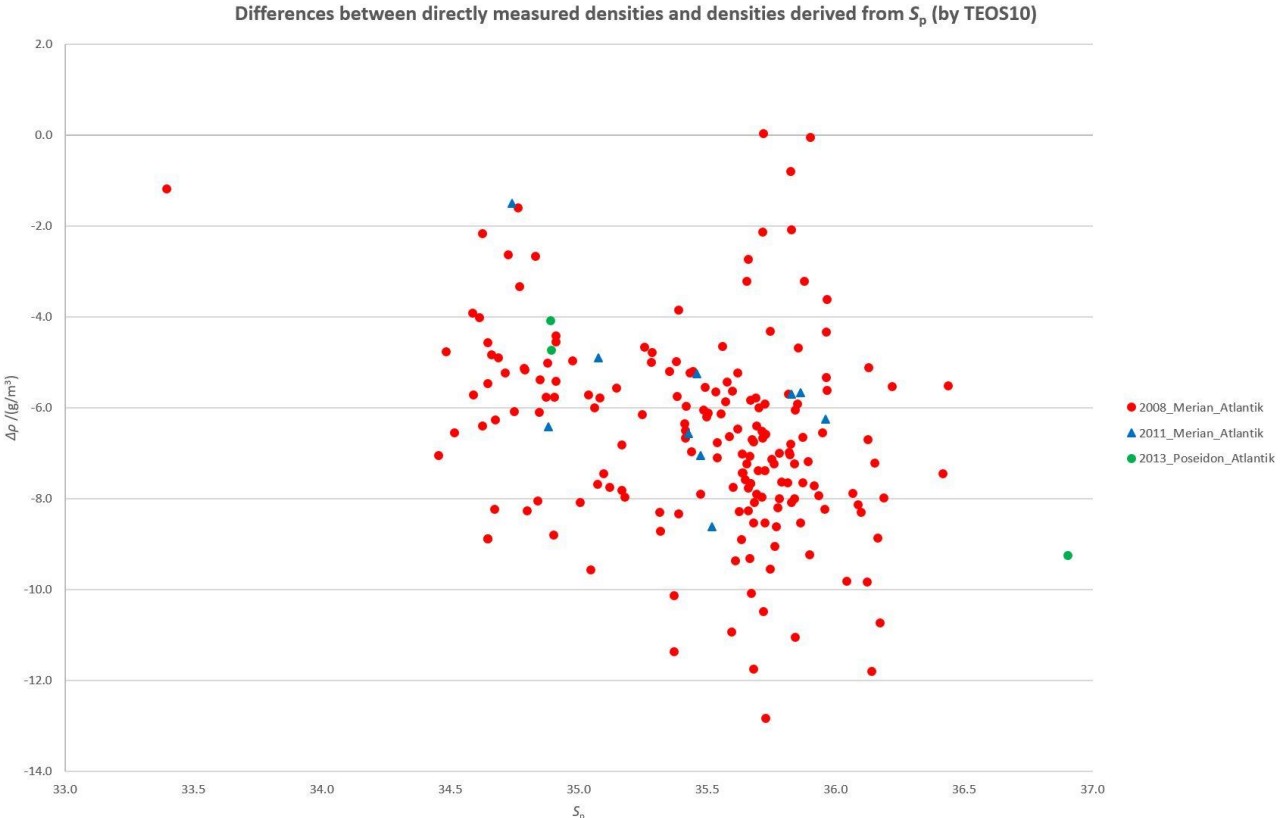

**Figure 3: Example for differences between directly measured densities and values calculated from TEOS-10 with measured Practical Salinities, $S_P$. Samples taken in 2008, 2011 and 2013 from eastern Atlantic surface waters between 33 °N and 18 °S show**
5  **systematic negative density anomalies up to 13 ppm (diagram courtesy Stefan Weinreben, IOW, priv. comm.).**





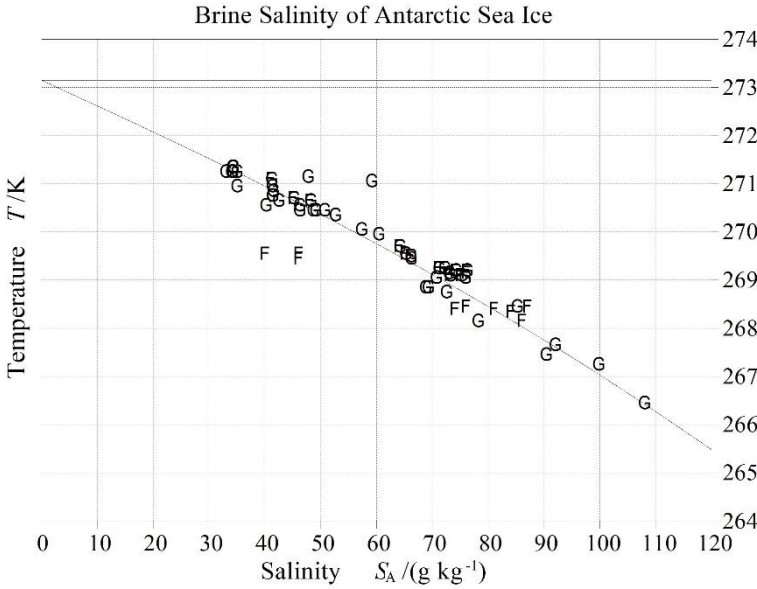

**Figure 4: Brine salinity computed from TEOS-10 as a function of temperature at atmospheric pressure, compared with measured results for brine pockets of Antarctic sea ice. Symbol "F": data of Fischer (2009), "G": data of Gleitz et al. (1995). As these observational data were not included in the construction of the 2008 Gibbs function, note that the curve is a theoretical prediction derived from the thermodynamic potentials of seawater and ice.**





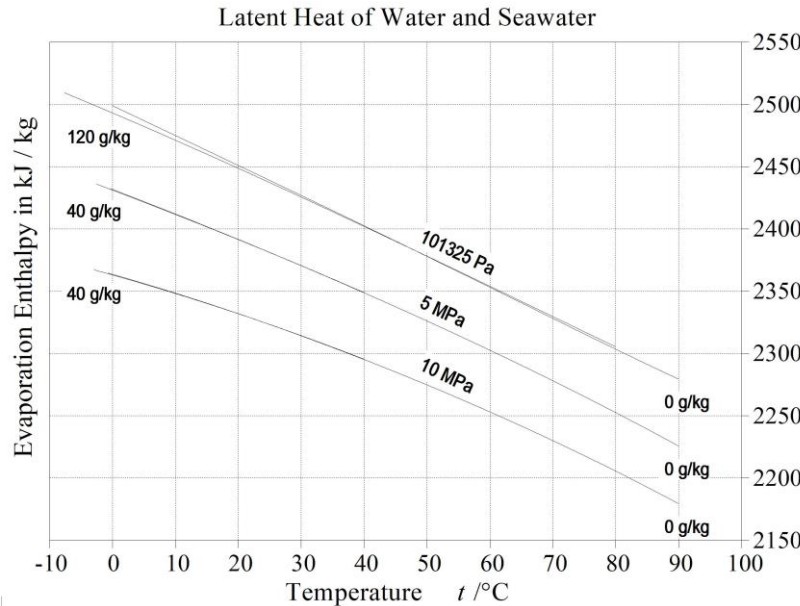

**Figure 5: Isobaric evaporation enthalpy of water, indicated by "0 g/kg", and of seawater with salinities 120 g kg$^{-1}$ at 101325 Pa and 40 g kg$^{-1}$ at 5 MPa and 10 MPa, as functions of the temperature, computed from TEOS-10. Here, the latent heat of seawater is derived from the total heat capacity of a 2-phase seawater-air composite, reduced by the two separate single-phase heat capacities involved. The salinity corrections to the latent heats of pure water are very small. At high pressure the validity of the Gibbs function of seawater is restricted to maxima of 40 g kg$^{-1}$ and 40 °C, but at atmospheric pressure it is valid up to 120 g kg$^{-1}$ and 80 °C. The lower temperature bounds shown are the particular freezing points of water or seawater (Feistel et al., 2010b).**



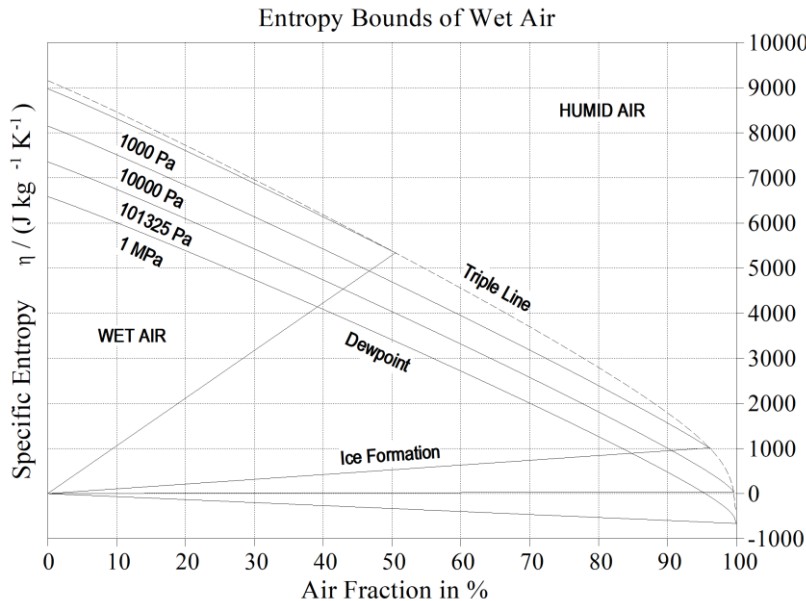

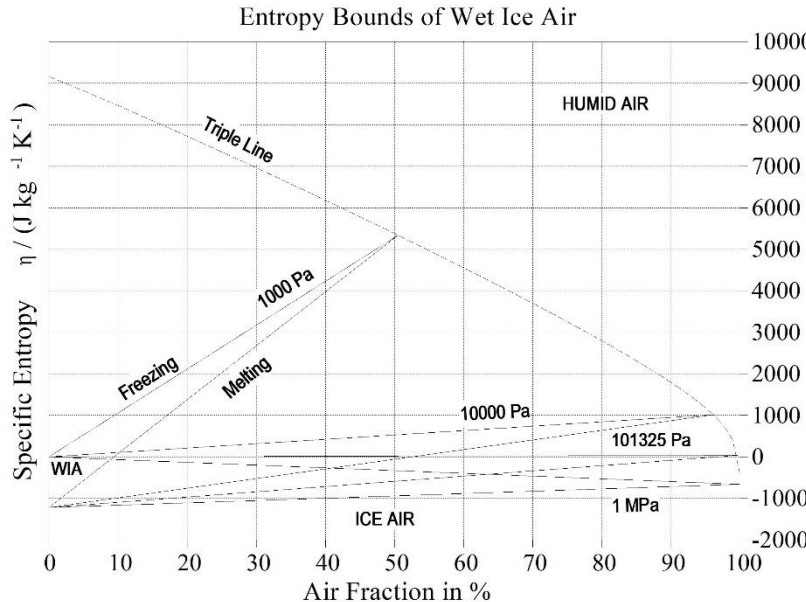

**Figure 6: Phase diagrams of composite air-water-ice systems ("clouds"), depending on the entropy and dry-air mass fraction of a given parcel, computed from TEOS-10 (Feistel et al., 2010a). Natural clouds with air fractions of 98-99 % at pressures between 20 000 and 101 325 Pa are located in the down-right corner of the diagrams. Metastable states, such as subcooled liquid droplets (WMO, 2008), are not considered here.**

**Upper panel: In the triangle-shaped, 2-phase regions "Wet Air", humid air is in equilibrium with liquid water. The tips of these wet-air wedges are triple points, located along the "Triple Line" depending on the pressure (or altitude) as indicated. Above the wedge, between the "Dewpoint" curve and "Triple Line", humid air is the only stable phase. Between "Dewpoint" and "Ice Formation", liquid water is in equilibrium with humid air.**



***Lower panel*:** **In the triangle-shaped, 3-phase regions "WIA" (wet ice air), humid air is in equilibrium with both liquid water and ice at the same time. The tips of theses WIA wedges are triple points, located along the "Triple Line" depending on the pressure (or altitude) as indicated. Above the wedge, the cloud contains only liquid water (wet air), if at all, while below the wedge, ice is the only condensed stable phase (ice air) in equilibrium with humid air.**

5    ***When adiabatically ascending*,** **entropy and dry-air fraction of an air parcel remain unchanged, even when passing the transition to cloud formation (the isentropic condensation level), and so does the parcel's representative point in the diagrams. With decreasing pressure, the "wet-air" and "wet-ice-air" regions will pan to the left, and states located initially between the "Dewpoint" and "Triple Line" get into the wet-air region first, then become wet ice air, and finally ice air. Points located above the "Triple Line" condensate to ice directly without passing any intermediate liquid phase. Using such diagrams, the parcel's phase changes during**

10    **convection may be predicted from its conservative properties at the ground.**





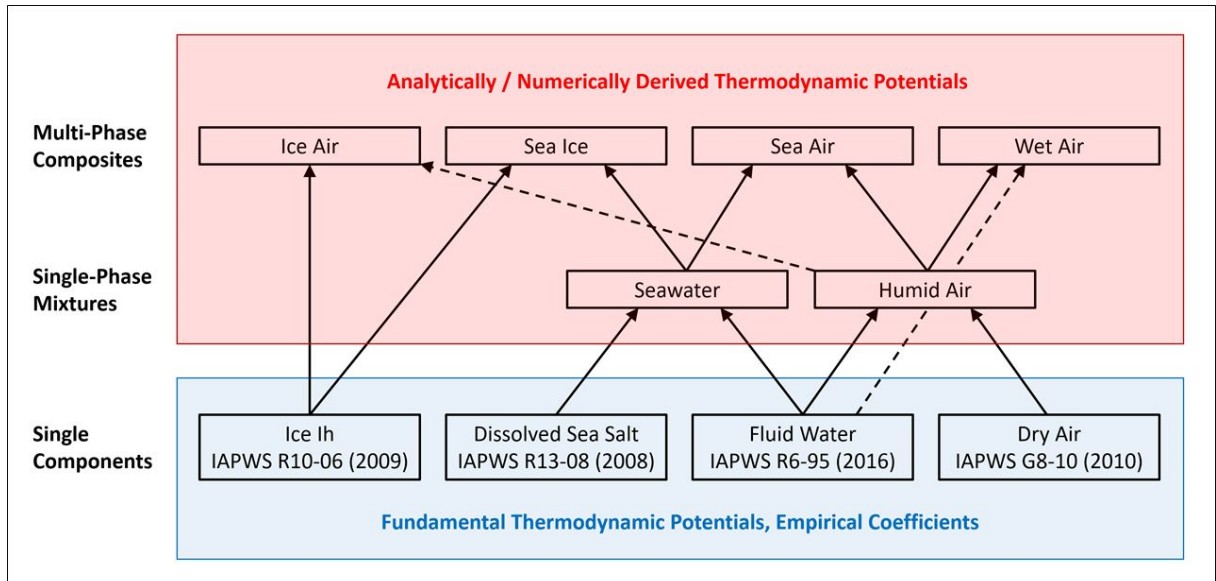

**Figure 7: Hierarchical structure of TEOS-10, based on four fundamental thermodynamic potentials defined in IAPWS documents for ice Ih, fluid water, dissolved sea salt, and dry air, including air-water interaction properties. Derived from those fundamental equations and their empirical coefficients by numerical and analytical mathematical methods, without introducing any additional empirical functions or coefficients, any thermodynamic properties can be computed for the pure substances (such as water and ice), their mixtures (such as seawater and humid air) and multi-phase composites (such as sea ice, ocean-atmosphere equilibria, and clouds).**



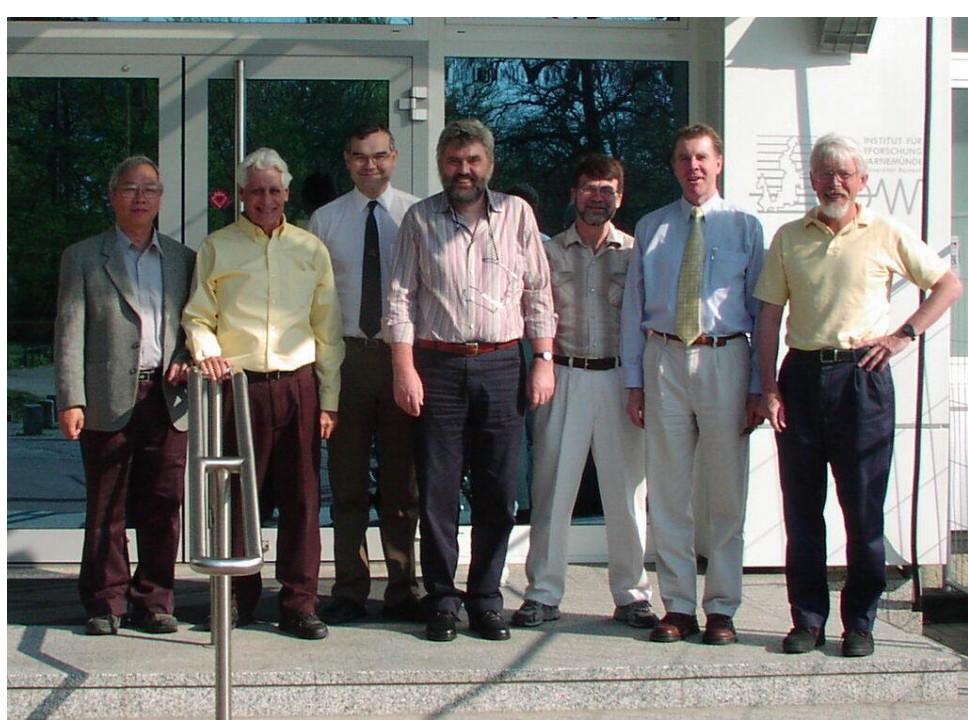

**Figure B.1: Participants from left to right: Chen-Tung Arthur Chen (Taiwan), Frank J. Millero (USA), Brian A. King (UK), Rainer Feistel (Germany), Daniel G. Wright (Canada), Trevor J. McDougall (Australia), Giles M. Marion (USA). Photo taken in front of the Baltic Sea Research Institute, IOW.**





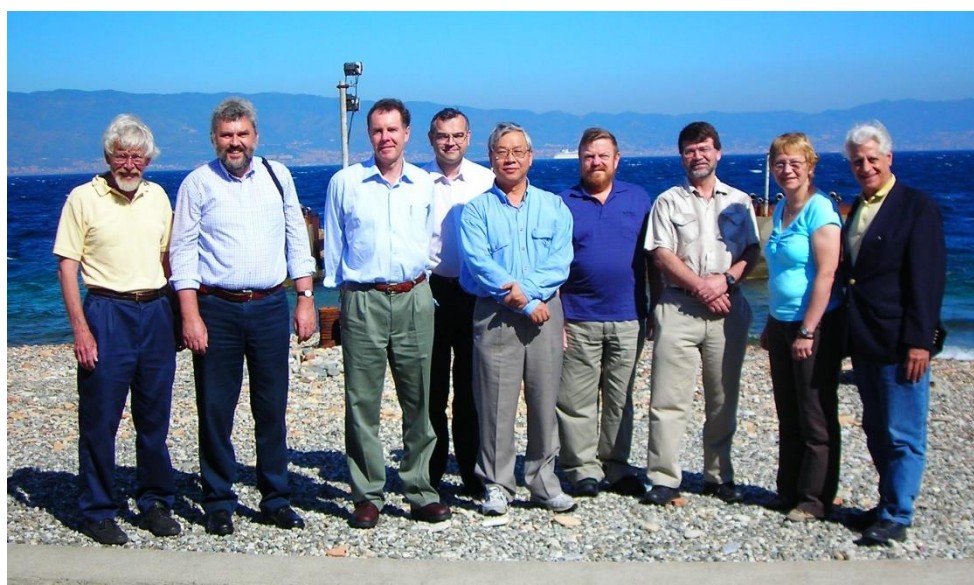

**Figure B.2: Participants from left to right: Giles M. Marion (USA), Rainer Feistel (Germany), Trevor J. McDougall (Australia), Brian A. King (UK), Chen-Tung Arthur Chen (Taiwan), David Jackett (Australia), Daniel G. Wright (Canada), Petra Spitzer (Germany), Frank J. Millero (USA). Photo taken at the shore of the Strait of Messina, with Sicily Island in the back.**



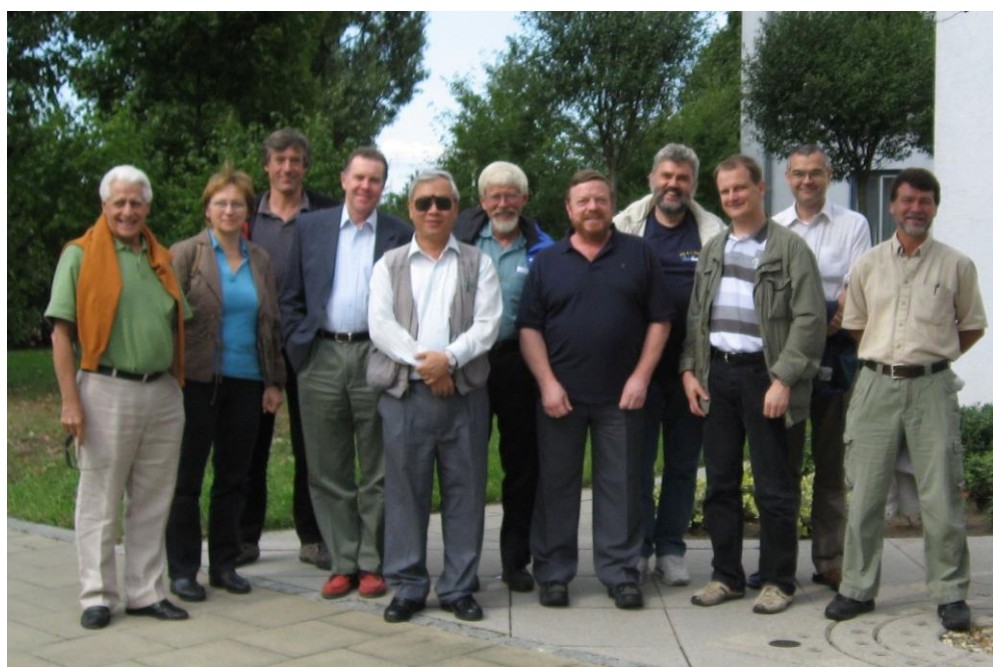

**Figure B.3: Participants from left to right: Frank J. Millero (USA), Petra Spitzer (Germany), Nigel Higgs (UK), Trevor J. McDougall (Australia), Chen-Tung Arthur Chen (Taiwan), Giles M. Marion (USA), David Jackett (Australia), Rainer Feistel (Germany), Steffen Seitz (Germany), Brian A. King (UK), Daniel G. Wright (Canada).**