# Peer review of "Thermodynamic Properties of Seawater, Ice and Humid Air: TEOS-10, Before and Beyond"

_Ocean Science, 2018_

## Referee Comment (RC1) · T. McDougall (Referee) · 19 Mar 2018

Review the paper "Thermodynamic Properties of Seawater, Ice and Humid Air: TEOS-10, Before and Beyond" by Rainer Feistel

This is a review article, on the occasion of Rainer winning the Fridtjof Nansen Medal of EGU, and delivering the Fridtjof Nansen address in Vienna. The article is very well written and it provides an invaluable insight into the workings of SCOR/IAPSO WG127 and the motivations of Rainer in addressing the many developments that were required to ultimately deliver the three thermodynamic potentials which are TEOS-10.

I have only very minor comments, as follows.

Line 202, delete "meanwhile"

Line 291. Should "Hexagonal ice I is" be "Hexagonal ice Ih is"

Line 297. Replace "wide range on temperatures" with "wide range of temperatures"

Line 365. I suggest replacing "controversial emotional" with "forthright"

Line 420. It would be clearer to replace "the Reference Salinity (9) is to corrected by" with "the Absolute Salinity is to estimated by"

Line 424. Replace "from density" with "using density"

Line 431. At 35 g/kg, T=300K and p = 0dbar, the saline contraction coefficient is about 0.72 not 0.66.

Line 433. Replace "received appreciation again for" with "relies on"

Line 617. Replace "latent heat of seawater" with "latent heat of evaporation of seawater"

Line 804. Replace "was commonly established" with "was jointly established"

Line 846. I suggest replacing "after many long and vivid discussions" with "after many long and memorable discussions"

Lines 947-948. Replace "BIPM has not recommended yet any RH definition by now." with "BIPM has not yet recommended any RH definition."

Line 1838. Replace "with Sicily Island in the back." with "with Sicily Island in the background."

---

## Short Comment (SC1) · 4 Apr 2018

**Short comment on ‚Thermodynamic properties of seawater, ice and humid air – TEOS-10, before and beyond' by R. Feistel**

H Schmidt

TEOS-10 is the step to theoretical consistency in the description of thermodynamic properties of humid air, seawater and ice. I have questions that are more or less related to accuracies.

*Page 6 Line 11*: The ITS-90 is an empirical temperature scale as described. The thermodynamic temperature $T$ therefore differs from the ITS-90 temperature $T_{90}$. Based on a request from the Consultive Committee for Thermometry (BIPM-CCT), Fischer et al. (2011)[1] gave updated data (and accuracies) of $\Delta T = T - T_{90}$. For 20 °C (and 30 °C), $\Delta T \approx 2$ mK (and $\Delta T \approx 4$ mK) with an accuracy of 0.8 mK.

In addition to the triple point of water, where no difference is expected in the new temperature scale, it may be interesting to know the expected $\Delta T$ for other temperatures, say 30 °C, as the 4 mK given by Fischer et al. are usually not considered in calculations.

*Page 7 Line 26–30*: Since the sound speed measurements of Del Grosso & Mader (1972)[2] and Chen & Millero (1977)[3] there has been a discussion, because the datasets have been inconsistent. Any correction or discussion on the measurements of Chen & Millero led to a correction towards the sound speeds of Del Grosso & Mader, especially the correction suggested by Millero & Li (1994)[4].

In developing TEOS-10, the sound speeds of Del Grosso & Mader (1972), i.e. those calculated by the equation of Del Grosso (1974)[5], were used instead of those of Chen & Millero (1977) or Millero & Li (1994) (Feistel, 2003[6], 2008[7]). Since Chen & Millero (1977) measured the sound speed in seawater relatively to those in water, there was the approach to replace the water sound speeds used by Chen & Millero by IAPWS-95 sound speeds. The result of this approach was summarized as (Feistel, 2003, p. 61):

"The new IAPWS95 sound speed formula suggested the hope that these problems with Chen-Millero sound speeds may now be eventually resolved in a natural way, but unfortunately this could not be achieved by a simple replacement of the pure water parts [..]."

However, in the article under discussion (p. see above):

"In TEOS-10, the IAPWS-95 equation replaced the earlier equations of state of liquid water [..]. This change of the pure-water equation made it possible to resolve systematic problems previously encountered with the sound speed of seawater at high pressures (Dushaw et al., 1993; Millero and Li, 1994; Feistel, 2003)."

What is meant by this second statement, as it somehow seems to contradict that first statement?

[1] Fischer J et al. (2011), Int J Thermophys, 32, 12-25, https://doi.org/10.1007/s10765-011-0922-1.
[2] Del Grosso VA & Mader CW (1972), J Acoust Soc Am, 52, 961-74, https://doi.org/10.1121/1.1913202.
[3] Chen C-T & Millero FJ (1977), J Acoust Soc Am, 62, 1129-30, https://doi.org/10.1121/1.381646.
[4] Millero FJ & Li X (1994), J Acoust Soc Am, 95, 2757-59, https://doi.org/10.1121/1.409844.
[5] Del Grosso VA (1974), J Acoust Soc Am, 56, 1084-91, https://doi.org/10.1121/1.1903388.
[6] Feistel R (2003), Progr Ocean, 58, 43-114, https://doi.org/10.1016/S0079-6611(03)00088-0.
[7] Feistel R (2008), Deep Sea Res I, 55, 1639-71, https://doi.org/10.1016/j.dsr.2008.07.004.

*Page 10 Line 27*: Figure 3 is introduced exemplifying the use of the salinity anomaly dSA. Figure 3 suggests a negative mean salinity anomaly of about -0.008g/kg for Atlantic surface water although TEOS-10 suggests a value of about 0.000g/kg for the region of interest (http://www.teos-10.org/pubs/gsw/pdf/SAAR.pdf, Figure 2).

What are the reasons for the significant negative anomaly shown in Figure 3?

*Page 12 Line 27*: "[..] SA is as accurate as SP [..]"

TEOS-10 uses the absolute salinity SA as input variable for calculations. However, SA cannot be measured directly in the ocean. Instead, the practical salinity SP is measured and converted to SA using the factor f=1.004715g/kg. For standard seawater it is assumed that SA matches the reference salinity SR. However, SR is based on measurements of standard seawater with an estimated accuracy of 0.014g/kg (Millero et al., 2008, p. 60)[8]. By contrast, practical salinity of standard seawater can be measured with an accuracy of 0.002 (=0.002g/kg) or reproduced even more accurately.

How can SR or SA be as accurate as SP?

*Page 19 Line 19–31*: Measurements of standard seawater density in addition to salinity „[..] could grant the requisite long-term stability of the SSW standard [..]"

SSW is essential in practical salinity measurement, as it cannot pe prepared artificially with the required accuracy nor stored without changes in its composition in the long term. Density measurement can detect changes in the standard seawater composition or preparation. It is possible to substitute the KCl solution in the preparation process to normalize standard seawater to S=35 with a significant loss in accuracy (0.0004 vs 0.003 in practical salinity).

How can density measurement grant long-term stability of standard seawater?
* * *
[8] Millero et al. (2008), Deep Sea Res I, 55, 50-72, https://doi.org/10.1016/j.dsr.2007.10.001.

---

## Author Comment (AC1) · 18 Apr 2018

**Rainer Feistel, 18 Apr 2018:**

**Reply to the short comments of Hannes Schmidt, 04 Apr 2018**

*Comment on Page 6 Line 11*: The ITS-90 is an empirical temperature scale as described. The thermodynamic temperature $T$ therefore differs from the ITS-90 temperature $T_{90}$. Based on a request from the Consultive Committee for Thermometry (BIPM-CCT), Fischer et al. (2011)[1] gave updated data (and accuracies) of $\Delta T = T - T_{90}$. For 20 °C (and 30 °C), $\Delta T \approx 2$ mK (and $\Delta T \approx 4$ mK) with an accuracy of 0.8 mK.
In addition to the triple point of water, where no difference is expected in the new temperature scale, it may be interesting to know the expected $\Delta T$ for other temperatures, say 30 °C, as the 4 mK given by Fischer et al. are usually not considered in calculations.

**Reply:** Please find more details on the new thermodynamic temperature scale at

- Gavioso RM et al. 2016, Progress towards the determination of thermodynamic temperature with ultra-low uncertainty. Phil. Trans. R. Soc. A 374: 20150046.
  http://dx.doi.org/10.1098/rsta.2015.0046
- Underwood R, de Podesta M, Sutton G, Stanger L, Rusby R, Harris P, Morantz P, Machin G. 2016 Estimates of the difference between thermodynamic temperature and the International Temperature Scale of 1990 in the range 118 K to 303 K. Phil. Trans. R. Soc. A 374: 20150048. http://dx.doi.org/10.1098/rsta.2015.0048

For higher temperatures, see also the CCT reports of

- Fellmuth et al., *Mise en Pratique* of the definition of the kelvin (MeP-K),
  https://www.bipm.org/cc/CCT/Allowed/28/CCT_28_MeP-K_cg-2017-32.pdf

and

- Fischer et al., WORKING GROUP FOR CONTACT THERMOMETRY REPORT TO CCT June 2017
  https://www.bipm.org/cc/CCT/Allowed/28/WG_CTh_report_2017-57.pdf

*Comment on Page 7 Line 26–30:* Since the sound speed measurements of Del Grosso & Mader (1972) and Chen & Millero (1977) there has been a discussion, because the datasets have been inconsistent. Any correction or discussion on the measurements of Chen & Millero led to a correction towards the sound speeds of Del Grosso & Mader, especially the correction suggested by Millero & Li (1994). In developing TEOS-10, the sound speeds of Del Grosso & Mader (1972), i.e. those calculated by the equation of Del Grosso (1974), were used instead of those of Chen & Millero (1977) or Millero & Li (1994) (Feistel, 2003, 2008). Since Chen & Millero (1977) measured the sound speed in seawater relatively to those in water, there was the approach to replace the water sound speeds used by Chen & Millero by IAPWS-95 sound speeds. The result of this approach was summarized as (Feistel, 2003, p. 61): "The new IAPWS95 sound speed formula suggested the hope that these problems with Chen-Millero sound speeds may now be eventually resolved in a natural way, but unfortunately this could not be achieved by a simple replacement of the pure water parts [..]." However, in the article under discussion (p. see above): "In TEOS-10, the IAPWS-95 equation replaced the earlier equations of state of liquid water [..]. This change of the pure-water equation made it possible to resolve systematic

problems previously encountered with the sound speed of seawater at high pressures (Dushaw et al., 1993; Millero and Li, 1994; Feistel, 2003)."

What is meant by this second statement, as it somehow seems to contradict that first statement?

**Reply:** In contrast to, say, heat capacity or specific volume, the sound speed expressed in terms of the Gibbs function is a complex nonlinear expression. So is not possible to simply "subtract" an obsolete "pure-water part" from seawater sound speed data and replace it by some improved values, like this was done with the Gibbs function when IAPWS-95 was specified as its pure-water part. As Millero and Li (1994) report, the problems encountered with Chen & Millero (1977) sound speeds resulted from the pure-water reference that Chen and Millero used during their measurements. It turned out that a provisional *a-posteriori* correction of that pure-water part by Millero and Li (1994) was not fully satisfactory because the required raw data are no longer available. Therefore, TEOS-10 has completely refrained from using Chen & Millero (1977) sound speed data and combined for the 2003 and 2008 Gibbs functions the Del Grosso (1974) sound speed with IAPWS-95 density for pure water and with seawater density and thermal expansion data as described in Feistel (2003). In the multi-property fit of the Gibbs function, this combination of data turned out to be mutually consistent and resolved the previous deviations found by Dushaw et al. (1993). So, while it appeared to be impossible to satisfactorily correct Chen & Millero (1976) data by IAPWS-95 (first statement), replacing those data by Del Grosso (1974) in the fit of the Gibbs function resolved the systematic problems encountered previously with the sound speed of seawater at high pressures (second statement).

***Comment on Page 10 Line 27:*** Figure 3 is introduced exemplifying the use of the salinity anomaly dSA. Figure 3 suggests a negative mean salinity anomaly of about -0.008g/kg for Atlantic surface water although TEOS-10 suggests a value of about 0.000g/kg for the region of interest (http://www.teos-10.org/pubs/gsw/pdf/SAAR.pdf, Figure 2).

What are the reasons for the significant negative anomaly shown in Figure 3?

**Reply:** Actually, answering this question belongs to current research tasks of JCS. Negative density anomalies of similar magnitude as those displayed have meanwhile been found in various regions of the world ocean, and even in some (but not all) certified SSW samples (see e.g. Fig. 6 in www.ocean-sci.net/6/3/2010/). So far, the reasons are elusive; working hypotheses may include unknown chemical (possibly organic?) composition anomalies, isotopic composition anomalies of water, previously unnoticed systematic deviations in the background data of TEOS-10, sample pollution such as by micro-plastic, or measurement errors of other unknown origin. Only after collection and analysis of more data and discovering responsible causes it can be discussed whether the gsw_SAAR library function may need to be updated regionally in the future.

***Comment on Page 12 Line 27:*** "[..] SA is as accurate as SP [..]"

TEOS-10 uses the absolute salinity SA as input variable for calculations. However, SA cannot be measured directly in the ocean. Instead, the practical salinity SP is measured and converted to SA using the factor f=1.004715g/kg. For standard seawater it is assumed that SA matches the reference salinity SR. However, SR is based on measurements of standard seawater with an estimated accuracy

of 0.014g/kg (Millero et al., 2008, p. 60)8. By contrast, practical salinity of standard seawater can be measured with an accuracy of 0.002 (=0.002g/kg) or reproduced even more accurately.

How can SR or SA be as accurate as SP?

**Reply:** While it is correct that SR is an estimate for the mass of dissolved sea salt in SSW with an estimated uncertainty of 7 mg/kg (Millero et al., 2008, p. 60, 70), the questioned sentence, however, begins with the clause "If defined by a fixed conversion factor for a reference composition, $S_A$ is as accurate as $S_P$". In fact, SR is related to SP by a fixed numerical factor, and this conversion between different salinity units of the input variable of the Gibbs function has no effect on the uncertainty of any derived results.

*Comment on Page 19 Line 19–31*: Measurements of standard seawater density in addition to salinity „[..] could grant the requisite long-term stability of the SSW standard [..]"

SSW is essential in practical salinity measurement, as it cannot pe prepared artificially with the required accuracy nor stored without changes in its composition in the long term. Density measurement can detect changes in the standard seawater composition or preparation. It is possible to substitute the KCl solution in the preparation process to normalize standard seawater to S=35 with a significant loss in accuracy (0.0004 vs 0.003 in practical salinity).

How can density measurement grant long-term stability of standard seawater?

**Reply:** This question is discussed in Metrologia 53 (2016) R12–R25, doi: 10.1088/0026-1394/53/1/R12: "A suggested new concept that takes advantage of currently available density measurement technology and at the same time leaves established oceanographic practice largely unaffected is a combination of conductance ratio and density measurement (Seitz et al 2011) [www.ocean-sci.net/7/45/2011/]. In this concept, the salinity of SSW samples can be additionally certified (or at least checked) by density measurements in combination with the TEOS-10 equation of state. At a given reference temperature and pressure, the density of an SSW sample corresponding to a specified salinity is measured by the sample's producer and its Practical Salinity value is calculated via the equation of state and the Reference Salinity, equation (5). This value can be compared directly with the value achieved according to the PSS-78 procedures, which will reveal possible longer-term changes in SSW properties. Recent investigations supported the validity of this procedure (EMRP 2010). However, many practical aspects must still be investigated before it would be feasible to transition to obtaining salinity from density instead of from conductance ratio."

So, the basic idea is to calibrate a CTD conductivity sensor with respect to the certified density of seawater reference samples. The sample density can be verified experimentally at any time against SI standards, independent of any prepared and possibly aging or varying artefacts. The calibration density is a measure of a 'density salinity' intended to be defined in the future (www.ocean-sci.net/7/1/2011/). While this concept does not grant IAPSO standard seawater to become "more stable" than currently, it offers a so-far unavailable option of measuring (and correcting for) suspected long-term (century scale) changes of the reference material against an ultimately stable metrological reference.

---

## Referee Comment (RC2) · R. Tailleux (Referee) · 23 Apr 2018

This well written paper surveys some of the theoretical basis underlying the new thermodynamic standard for seawater TEOS-10, while also providing some discussion of remaining issues to be tackled in the future. One of the most striking aspect of TEOS-10 is its dramatic departure from the theory and concepts of the previous standard. While there is no doubt that the new standard represents a major advance over the previous standard, the unfamiliar character of several of its new concepts has also resulted in many oceanographers struggling to grapple with what the new framework actually means and what its actual implications for oceanographic practice and future developments are. This review — but this is also true of the TEOS-10 manual — makes occasional statements and assertions that are by no means self-evident; on the very

few occasions where this occur, it would therefore be helpful if the author could attempt to be more pedagogical and provide more details on the theoretical justification for some of the most intriguing aspects of the TEOS-10 listed below.

1. Page 10, Line 21-22: **TEOS-10 is the first international seawater where chemical composition anomalies are explicitly accounted for.** This part of the review is very unsatisfactory (but then this is a problem of the TEOS-10 manual as well), as it is not possible from the information given to understand how exactly TEOS-10 account for chemical composition anomalies, nor what is the underlying theoretical justification for it. Indeed, TEOS-10 is presented as providing for the first time a synthesis of all possible thermodynamic information about reference composition standard seawater by means of a master thermodynamic potential (the Gibbs function). The statement 'TEOS-10 accounts for the first time for chemical composition anomalies' suggests therefore that TEOS-10 provides a mechanism for quantifying the impact of composition anomalies on all possible thermodynamic functions, but all what is explicitly discussed is density, which is only one of the many thermodynamic quantities of interest. Moreover, saying that TEOS-10 can account for chemical composition anomalies suggest that it is in principle possible to deduce how all possible thermodynamic quantities are affected by composition anomalies. What the author discusses here, however, is how to compute Absolute Salinity from the knowledge of density, which seems to be the opposite of what is needed. What is the theoretical basis for believing that thermodynamic quantities are only determined by the total mass fraction of the dissolved components? Why is this not discussed in the TEOS-10 manual nor in the present review? Is the Absolute Salinity $S_A = S_R + \delta S_A$ really a single variable, or is it really two more more physical variables? How can we derive a mathematically well-posed problem for $S_A$ in that case? Don't we need an equation for both $S_R$ and $\delta S_A$? And is it really possible to derive an evolution equation for $\delta S_A$ in terms of a single evolution equation or do we actually several evolu-

**OSD**

[Figure]

tion equations? Could the author also explain what is Millero's rule exactly, and how is it possible to investigate its validity? Could the author also comment on the possible use of FREZCHEM to construct a Gibbs function for seawater as a function of more than just one composition variable?

2. Page 12, Line 29. **All physical, chemical and oceanographic, theoretical as well as numerical models do actually rely on $S_A$ rather than $S_p$. Outside oceanography is the only the scientific community recognises salinity.**. I don't really understand these statements. What does the author mean by 'Absolute Salinity'? Does he mean density salinity or Reference Composition salinity? Or is the author using the term 'Absolute Salinity' as a generic way to refer to a quantity expressed in standard composition units such as g/kg? For standard seawater, $S_p$ is mathematically equivalent to $S_R$, since the two are related by a fixed conversion factor, so in some sense, the distinction between $S_p$ and Absolute Salinity is only justified for seawater that differs from standard seawater. It seems to me, however, that from a practical viewpoint, one does not really have the choice at the moment when numerically modelling the ocean but to neglect composition anomalies and to assume fixed composition, since if only one evolution equation is used to describe salinity, it has to be for reference composition salinity $S_R$ or equivalently $S_p$, since the equations for both quantities are exactly the same but for boundary conditions.

3. Page 19. I find the issues pertaining to SI traceability quite tricky to understand, and I believe that many readers would appreciate a more pedagogical treatment here. To the extent that density of seawater may also be affected by such effects as dissolved $CO_2$, air bubbles, microplastic, etc..., which may affect density without contributing to the mass of dissolved tracers, it is unclear to what extent density salinity is always a good proxy for Absolute Salinity? If one cannot be sure that density salinity is a good enough proxy for Absolute Salinity, how confident can we be that 'seawater density is the only promising candidate for

SI-traceability measurements in the oceanographic practice?' Moreover, isn't it a problem that density also depends on pressure and on the precise value of gravity at the place of measurement?

---

## Author Comment (AC3) · 28 Apr 2018

Thanks to the referee for his detailed questions. Expecting similar questions to appear also with other readers, the salinity parts of the ms have been revised (supplement attached) by including additional text parts and references, addressing the issues raised by the referee.

Please also note the supplement to this comment:
https://www.ocean-sci-discuss.net/os-2018-19/os-2018-19-AC3-supplement.pdf
* * *
[Figure]

**Supplement:**

[revised manuscript text omitted]

---

## Author Response (AR1)

Title: **Thermodynamic Properties of Seawater, Ice and Humid Air: TEOS-10, Before and Beyond**
Author(s): Rainer Feistel
MS No.: os-2018-19
MS Type: Review article
Iteration: Revised Submission

**Detailed point-by-point response to all referee comments**

In the following, for easy distinction, referee comments are on gray background, and author responses on white paper. Text in yellow indicates changes made to the submission in response to referee 1, and text in blue to referee 2.

**Referee 1:**

Thanks for the suggestions given.

This is a review article, on the occasion of Rainer winning the Fridtjof Nansen Medal of EGU, and delivering the Fridtjof Nansen address in Vienna. The article is very well written and it provides an invaluable insight into the workings of SCOR/IAPSO WG127 and the motivations of Rainer in addressing the many developments that were required to ultimately deliver the three thermodynamic potentials which are TEOS-10.

I have only very minor comments, as follows.
Line 202, delete "meanwhile"
- done
Line 291. Should "Hexagonal ice I is" be "Hexagonal ice Ih is"
- sentence changed to:
  Hexagonal ice I, or *ice Ih*, is the only stable solid phase of water in the terrestrial atmosphere, hydrosphere and cryosphere (in contrast to metastable cubic ice I, or *ice Ic*, which appears temporarily in cloud formation processes).
Line 297. Replace "wide range on temperatures" with "wide range of temperatures"
- done
Line 365. I suggest replacing "controversial emotional" with "forthright"
- done
Line 420. It would be clearer to replace "the Reference Salinity (9) is to corrected by" with "the Absolute Salinity is to estimated by"
- done
Line 424. Replace "from density" with "using density"
- done
Line 431. At 35 g/kg, T=300K and p = 0dbar, the saline contraction coefficient is about 0.72 not 0.66.
- sentence changed to:
  Here, $\beta$ is the haline contraction coefficient of seawater which has a typical value of about 0.73 at 35 g kg$^{-1}$, 1000 hPa and 300 K.
Line 433. Replace "received appreciation again for" with "relies on"
- done
Line 617. Replace "latent heat of seawater" with "latent heat of evaporation of seawater"
- done
Line 804. Replace "was commonly established" with "was jointly established"
- done
Line 846. I suggest replacing "after many long and vivid discussions" with "after many long and memorable discussions"
- done

Lines 947-948. Replace "BIPM has not recommended yet any RH definition by now." with "BIPM has not yet recommended any RH definition."
-   done

Line 1838. Replace "with Sicily Island in the back." with "with Sicily Island in the background."
-   done

**Referee 2:**

Thanks for the suggestions given.

This well written paper surveys some of the theoretical basis underlying the new thermodynamic standard for seawater TEOS-10, while also providing some discussion of remaining issues to be tackled in the future. One of the most striking aspect of TEOS-10 is its dramatic departure from the theory and concepts of the previous standard. While there is no doubt that the new standard represents a major advance over the previous standard, the unfamiliar character of several of its new concepts has also resulted in many oceanographers struggling to grapple with what the new framework actually means and what its actual implications for oceanographic practice and future developments are. This review—but this is also true of the TEOS-10 manual—makes occasional statements and assertions that are by no means self-evident; on the very few occasions where this occur, it would therefore be helpful if the author could attempt to be more pedagogical and provide more details on the theoretical justification for some of the most intriguing aspects of the TEOS-10 listed below.

1. Page 10, Line 21-22: **TEOS-10 is the first international seawater where chemical composition anomalies are explicitly accounted for.** This part of the review is very unsatisfactory (but then this is a problem of the TEOS-10 manual as well), as it is not possible from the information given to understand how exactly TEOS-10 account for chemical composition anomalies, nor what is the underlying theoretical justification for it. Indeed, TEOS-10 is presented as providing for the first time a synthesis of all possible thermodynamic information about reference composition standard seawater by means of a master thermodynamic potential (the Gibbs function). The statement 'TEOS-10 accounts for the first time for chemical composition anomalies' suggests therefore that TEOS-10 provides a mechanism for quantifying the impact of composition anomalies on all possible thermodynamic functions, but all what is explicitly discussed is density, which is only one of the many thermodynamic quantities of interest. Moreover, saying that TEOS-10 can account for chemical composition anomalies suggest that it is in principle possible to deduce how all possible thermodynamic quantities are affected by composition anomalies. What the author discusses here, however, is how to compute Absolute Salinity from the knowledge of density, which seems to be the opposite of what is needed. What is the theoretical basis for believing that thermodynamic quantities are only determined by the total mass fraction of the dissolved components? Why is this not discussed in the TEOS-10 manual nor in the present review?
-   A longer text part has been added to Section 3.4, explaining the basis and consequences of the so-called "Millero's Rule"

Is the Absolute Salinity SA = SR + δSA really a single variable, or is it really two more more physical variables? How can we derive a mathematically well-posed problem for SA in that case? Don't we need an equation for both SR and δSA? And is it really possible to derive an evolution equation for δSA in terms of a single evolution equation or do we actually several evolution equations?
-   TEOS-10 supports two independent salinity variables, SA and SR, rather than the single one, SP, of the previous standard. To the present knowledge, those two salinity variables appear practically sufficient for the description of most regionally observed anomalous seawater properties. Appendix A.20 of the TEOS-10 Manual investigates this question more deeply. For the resolution of more composition details affecting the Gibbs function, the related

fundamental physical-chemical properties of mixed, concentrated electrolyte solutions are only insufficiently well known yet, such as by Pitzer equations or other empirical formulas. An explaining text part has been added to Section 3.4

Could the author also explain what is Millero's rule exactly, and how is it possible to investigate its validity?
-   A longer text part has been added to Section 3.4, explaining the basis and consequences of the so-called "Millero Rule". See the references given there.

Could the author also comment on the possible use of FREZCHEM to construct a Gibbs function for seawater as a function of more than just one composition variable?
-   A paragraph has been added, briefly explaining some results obtained from Pitzer models that are capable of resolving the chemical composition. The related equations are relatively complicated (Feistel and Marion, 2007; Feistel et al., 2010d), and so far the accuracy of certain resulting properties, such as sound speeds, is insufficient for oceanographic use.

2. Page 12, Line 29. **All physical, chemical and oceanographic, theoretical as well as numerical models do actually rely on** $S_A$ **rather than** $S_P$**. Outside oceanography is the only the scientific community recognises salinity**. I don't really understand these statements. What does the author mean by 'Absolute Salinity'?
- Section 3.4 says that "TEOS-10 is formulated in terms of *Absolute Salinity*, $S_A$, defined as the mass fraction of sea salt dissolved in water". For the reader's convenience, the definition is repeated again in different words in Section 3.5. Outside oceanography, such as in solution chemistry or electrolyte theory, "Practical Salinity" is an unknown quantity, see e.g. Gamsjäger et al., GLOSSARY OF TERMS RELATED TO SOLUBILITY, Pure Appl. Chem., 80, 233–276, 2008. The IUPAC Gold Book does not even recognize the term "salinity". In contrast, "solute mass fraction" is a common quantity in thermodynamics of liquid mixtures. "Absolute Salinity" is the traditional oceanographic name for the solute mass fraction in seawater.

Does he mean density salinity or Reference Composition salinity?
-   Neither nor. Density Salinity is recommended as the currently best estimate for Absolute Salinity, while Reference-Composition Salinity is the currently best estimate for Absolute Salinity of SSW.

Or is the author using the term 'Absolute Salinity' as a generic way to refer to a quantity expressed in standard composition units such as g/kg?
-   No, there exist various salinity measures in g/kg (or similarly, in % or ‰) that do not match Absolute Salinity

For standard seawater, $S_P$ is mathematically equivalent to $S_R$, since the two are related by a fixed conversion factor, so in some sense, the distinction between $S_P$ and Absolute Salinity is only justified for seawater that differs from standard seawater. It seems to me, however, that from a practical viewpoint, one does not really have the choice at the moment when numerically modelling the ocean but to neglect composition anomalies and to assume fixed composition, since if only one evolution equation is used to describe salinity, it has to be for reference composition salinity $S_R$ or equivalently $S_P$, since the equations for both quantities are exactly the same but for boundary conditions.
-   Appendix A.20 of the TEOS-10 Manual discusses the question of modelling several salinity variables.

3. Page 19. I find the issues pertaining to SI traceability quite tricky to understand, and I believe that many readers would appreciate a more pedagogical treatment here. To the extent that density of seawater may also be affected by such effects as dissolved $CO_2$, air bubbles, microplastic, etc...,

which may affect density without contributing to the mass of dissolved tracers, it is unclear to what extent density salinity is always a good proxy for Absolute Salinity?

- Answering these questions is among the ongoing research tasks of JCS; some text has been added to section 5.1, raising certain aspects.

If one cannot be sure that density salinity is a good enough proxy for Absolute Salinity, how confident can we be that 'seawater density is the only promising candidate for SI-traceability measurements in the oceanographic practice?'

- This conclusion was drawn by WG127 in 2008 after long and detailed discussions. In the past 10 years, no better option has come up. It remains an open question whether this may change in some near future by new inventions or measuring techniques, unknown today. To Section 5.1, a sentence has been added that "No other seawater property has been demonstrated so far to be practically measurable with requisite accuracy and to be traceable to the SI at the same time."

Moreover, isn't it a problem that density also depends on pressure and on the precise value of gravity at the place of measurement?

- Density measurements are carried out under laboratory conditions at well-known air pressure. Using the TEOS-10 equation of state, the results are transferred to in-situ conditions at known pressure. This is briefly mentioned in additional text in Section 5.1. Gravity conditions affect the local pressure via the hydrostatic equation but are largely irrelevant if the pressure itself is measured rather than computed.

**Revised manuscript with all changes marked in either yellow or blue with respect to the originally submitted version**

[revised manuscript text omitted]

---

## Author Response (AR2)

**Topic Editor Decision: Publish subject to technical corrections** (23 May 2018) by Mario Hoppema

Comments to the Author:

Dear Dr. Feistel,

Thank you for your interesting manuscript and congratulation to the 2018 Fridtjof Nansen medal.

The two referees were happy with the manuscript and had only few comments. Your answers to those are satisfactory, however, I would have expected to see (part of) those back in the manuscript, as this would enhance the understanding for the readers. I am adding some comments, which I hope you will include in the final submission.

"Outside oceanography is the only the scientific community recognises salinity." I think the problem of the referee lies in the phrasing. Please consider: Oceanography is the only the scientific community that recognises salinity.

RF: text modified.

Referee #2: Does he mean density salinity or Reference Composition salinity?

\- Neither nor. Density Salinity is recommended as the currently best estimate for Absolute Salinity, while Reference-Composition Salinity is the currently best estimate for Absolute Salinity of SSW.

Editor: Did you address this in the manuscript?

RF: statement added to 3.4

Referee #2: For standard seawater, $S_P$ is mathematically equivalent to $S_R$, since the two are related by a fixed conversion factor, so in some sense, the distinction between $S_P$ and Absolute Salinity is only justified for seawater that differs from standard seawater. It seems to me, however, that from a practical viewpoint, one does not really have the choice at the moment when numerically modelling the ocean but to neglect composition anomalies and to assume fixed composition, since if only one evolution equation is used to describe salinity, it has to be for reference composition salinity $S_R$ or equivalently $S_P$, since the equations for both quantities are exactly the same but for boundary conditions.

- Appendix A.20 of the TEOS-10 Manual discusses the question of modelling several salinity variables.

Editor: Please briefly address this in the manuscript.

RF: This is already done in the last but one paragraph of section 3.4, but another sentence has been added there

Moreover, isn't it a problem that density also depends on pressure and on the precise value of gravity at the place of measurement?

\- Density measurements are carried out under laboratory conditions at well-known air pressure. Using the TEOS-10 equation of state, the results are transferred to in-situ conditions at known pressure. This is briefly mentioned in additional text in Section 5.1. Gravity conditions affect the local pressure via the hydrostatic equation but are largely irrelevant if the pressure itself is measured rather than computed.
Editor: Please briefly address this in the manuscript.

RF: Text added at the end of 5.1. Now that gravity is considered, another gravity aspect of ocean thermodynamics is added to 3.5.

P17, L13 I do not understand this sentence. Please make clear.

RF: Some explaining text added.

Reference Budéus update:

G. Th. Budéus: Potential bias in TEOS10 density of sea water samples,
Deep Sea Research Part I: Oceanographic Research Papers, 134, 41-47, 2018,
https://doi.org/10.1016/j.dsr.2018.02.005.

RF: updated, thanks

Burchard et al: update avauilable? Doi?

RF: doi updated (proofs have just arrived)

Feistel 2017. Please provide translation of the German title.

RF: title translation added […]

Comment by Hannes Schmidt: Comment on Page 7 Line 26–30
I think Schmidt touches a problem that needs more explanation in the manuscript. Please add (part of) your answer to the manuscript.
Comment by Hannes Schmidt: Comment on Page 10 Line 27
Same as above.

RF: related text has been added to 5.1

With best wishes
Mario Hoppema

[revised manuscript text omitted]